# GICDM: Mitigating Hubness for Reliable Distance-Based Generative Model Evaluation

**Nicolas Salvy** [1 2 3]   **Hugues Talbot** [2 1 3]   **Bertrand Thirion** [1 4 3]

## Abstract

Generative model evaluation commonly relies on high-dimensional embedding spaces to compute distances between samples. We show that dataset representations in these spaces are affected by the hubness phenomenon, which distorts nearest-neighbor relationships and biases distance-based metrics. Building on the classical *Iterative* Contextual Dissimilarity Measure (ICDM), we introduce Generative ICDM (GICDM), a method to correct neighborhood estimation for both real and generated data. We introduce a multi-scale extension to improve empirical behavior. Extensive experiments on synthetic and real benchmarks demonstrate that GICDM resolves hubness-induced failures, restores reliable metric behavior, and improves alignment with human assessment.

## 1. Introduction

Generative models have achieved significant progress in recent years, enabling the synthesis of data for arbitrary modalities, including critical areas such as medical imaging (Pinaya et al., 2022; Koetzier et al., 2024; Bluethgen et al., 2024). Evaluating the quality of generated data is essential to ensure its reliability for downstream applications.

Yet, structured data, such as images, are high-dimensional, which makes direct density estimation infeasible. A common approach for evaluating generative models is to use *simple* distributional approximations. For example, Fréchet Inception Distance (FID) variants (Heusel et al., 2017; Stein et al., 2023) assume normality of the distribution. This enables evaluation via a single score, typically the distance between the real and synthetic approximated distributions. However, beyond the untested assumptions, this aggregate

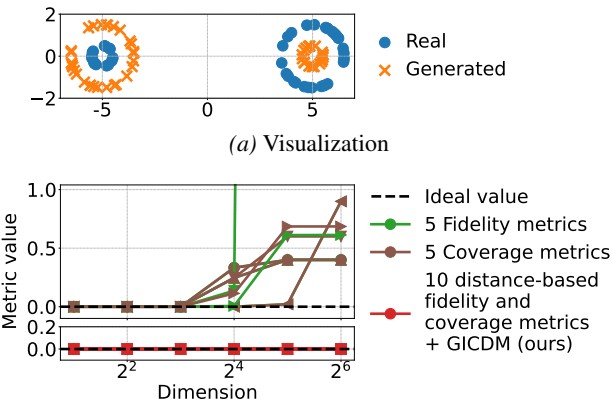

*(a)* Visualization

*(b)* Corresponding metric scores

*Figure 1.* **High dimension challenges distance-based metrics**: The scenario in *(a)* consists of real samples uniformly drawn from a 60/40 mixture of two hyperspheres, and generated samples uniformly drawn from a mixture with swapped radii and proportions. Standard metrics for this scenario are plotted in *(b)* as the dimension increases. The real and generated sets are disjoint, so all metrics should score 0; however, none of the standard metrics (top) do in high dimensions due to hubness: on the right side generated points spuriously appear closer to real points than real points are to each other. After applying GICDM (bottom), their scores correctly remain at 0. See Appendix I for individual metric results.

score makes it difficult to diagnose whether a generative model lacks realism or diversity (Sajjadi et al., 2018).

Pairs of fidelity and coverage metrics aim to measure these two aspects separately, typically using distances (Sajjadi et al., 2018; Naeem et al., 2020; Salvy et al., 2026). A synthetic sample is then considered realistic (high fidelity) if it is sufficiently close to real data, while a real point is considered covered if there are synthetic samples sufficiently close to it. The closeness threshold is usually defined locally as the distance from each real point to its $k$-th nearest real neighbor.

A recent position paper argued that all existing fidelity and diversity metrics are flawed (Räisä et al., 2025). This paper introduced a synthetic benchmark of tests for evaluating generative model metrics and showed that all current metrics fail to meet at least 40% of the success criteria. While subsequent work has led to improvements (Salvy et al., 2026),

[1]Inria, Palaiseau, France [2]CentraleSupélec, Gif-sur-Yvette, France [3]Université Paris-Saclay, Gif-sur-Yvette, France [4]CEA, Gif-sur-Yvette, France. Correspondence to: Nicolas Salvy <nicolas.salvy@inria.fr>.

*Proceedings of the 43rd International Conference on Machine Learning*, Seoul, South Korea. PMLR 306, 2026. Copyright 2026 by the author(s).

many failures persist. We argue that the underlying cause of many reported failures is related to high-dimensionality, specifically the *hubness* phenomenon. As shown in Figure 1, all metrics incorrectly yield non-zero scores for two disjoint uniform distributions on mixtures of hyperspheres in high dimensions. This happens because hubness distorts distances, making generated points spuriously appear closer to real points than real points are to each other.

Distance-based evaluation assumes that feature space distances are meaningful: close points should be structurally and semantically similar, and distant points should be structurally or semantically different. In practice, evaluation is performed in pre-trained embedding spaces, as feature extractors provide richer semantic representations than the raw observation space. However, modern embedding spaces are usually high-dimensional (e.g., 2048 for InceptionV3 (Szegedy et al., 2016), 4096 for DINOv3 (Siméoni et al., 2025), 1024 for CLAP (Elizalde et al., 2023)), making them vulnerable to the curse of dimensionality (Bellman, 1961).

A key aspect of this curse, known as *hubness* (Radovanovic et al., 2010), undermines the reliability of nearest-neighbor relationships in high-dimensional spaces (Beyer et al., 1999; Aggarwal et al., 2001). Specifically, certain points, called *hubs*, appear disproportionately often among the $k$-nearest-neighbors of other data points, even when they are semantically unrelated (Pachet & Aucouturier, 2004). At the same time, many other points, called *antihubs*, never appear as nearest-neighbors, meaning they effectively vanish from distance-based evaluation. This phenomenon arises from the structure of high-dimensional data distributions and is not simply due to limited sample size (Radovanovic et al., 2010).

Hubness has been recognized as problematic in various domains such as image recognition (Tomasev et al., 2011) and recommender systems (Hara et al., 2015). We show that hubness also affects modern embedding spaces (Table 4).

To enable reliable distance-based generative model evaluation in high-dimensional spaces, we aim to mitigate hubness while preserving metric validity. Our goals are: (1) to reduce hubness in the real dataset so that nearest-neighbor relationships are meaningful and symmetric, (2) to preserve the relative positioning of generated points with respect to real data, and (3) to ensure that the evaluation of each generated point is independent of the others, meaning its assessment depends solely on the real data. Standard hubness mitigation methods focus only on in-sample reduction and do not directly satisfy these requirements (Feldbauer & Flexer, 2019).

In this paper, we introduce GICDM, a hubness reduction method tailored for distance-based generative model evaluation. Our contributions are:

- **Demonstrating hubness**: We show that common embedding spaces for generative model evaluation exhibit hubness, and find that the *Iterative* Contextual Dissimilarity Measure (ICDM) (Jégou et al., 2010) is effective at reducing hubness in real datasets.

- **Hubness mitigation for generative model evaluation**: We show that hubness has a major impact on distance-based fidelity and coverage metrics. We adapt the ICDM method for evaluating generative models and enhance it with a careful filtering strategy. The result is GICDM, a hubness reduction method for pairs of real and generated sets.

This paper is organized as follows. Section 2 reviews the hubness phenomenon; Section 3 surveys hubness reduction methods; Section 4 discusses related evaluation metrics; Section 5 introduces GICDM; Section 6 presents experimental results; and Section 7 concludes.

## 2. Background: Hubness in High-Dimensional Distributions

The $k$-*occurrence* of a point $x \in D$, denoted $O_k(x)$, is the number of times $x$ appears among the $k$-nearest-neighbors of other points in the dataset $D$: $O_k(x) = \#\{y \in D \setminus \{x\} \mid x \in \mathcal{N}_k(y)\}$, where $\mathcal{N}_k(y)$ is the set of $k$-nearest-neighbors of $y$ in $D$. As dimensionality increases, the distribution of $k$-occurrences becomes increasingly skewed to the right, with the emergence of *hubs*: points with unusually high $O_k$ values. Conversely, this skew leaves a massive number of points with $O_k = 0$, known as *antihubs* (see Figure 2). This phenomenon is called *hubness* (Radovanovic et al., 2010). Unlike in low dimensions, where nearest-neighbor relationships are often reciprocal, hubness makes these relationships highly asymmetric, rendering the notion of "neighborhood" unreliable for many data points (Feldbauer et al., 2018; Feldbauer & Flexer, 2019).

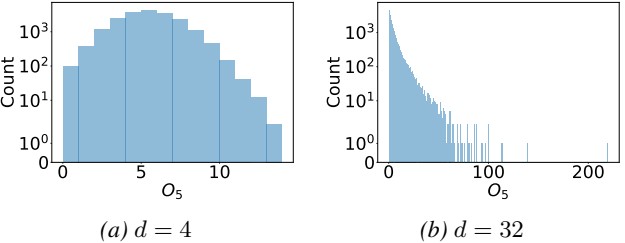

*Figure 2.* **Distribution of $k$-occurrences** ($k = 5$) for 20000 samples from a standard Gaussian in *(a)* $d = 4$ and *(b)* $d = 32$. The y-axis shows the count (log scale) for each $k$-occurrence value (x-axis). As dimension increases, the distribution skews right, leading to the emergence of *hubs* (frequent nearest-neighbors, high $O_k$) and *antihubs* (never neighbors, $O_k = 0$).

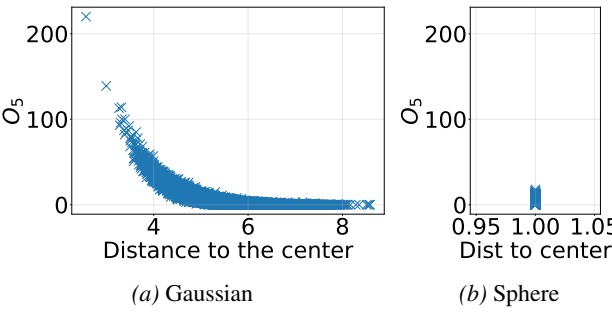

*(a) Gaussian*      *(b) Sphere*

*Figure 3.* **5-Occurrence vs Centrality** ($N = 20000$, $d = 32$): *(a)* (adapted from Radovanovic et al. (2010), Fig. 3) Standard Gaussian: hubness aligns with centrality; central points have higher occurrences. *(b)* Uniform sphere: removing centrality eliminates hubness. So, high dimensionality alone does not cause hubness.

## 2.1. Gaussian case

This phenomenon can be intuitively understood in the Gaussian case. Let $X, Y \sim \mathcal{N}(0, I_d)$ be independent standard Gaussian random vectors in $\mathbb{R}^d$.

**Spherical shell.** The squared norm $\|X\|^2$ follows a chi-squared distribution with $d$ degrees of freedom, $\|X\|^2 \sim \chi^2(d)$, with mean $d$ and standard deviation $\sqrt{2d}$. Therefore, in high dimensions, standard Gaussian samples concentrate on a thin spherical shell of radius $\sqrt{d}$ and thickness $O(d^{1/4})$ (Vershynin, 2018). See Figure 7 (top) for an illustration.

**Orthogonality.** Two independent high-dimensional Gaussian vectors tend to be nearly orthogonal (Vershynin, 2018). The inner product $X \cdot Y = \sum_{i=1}^{d} X_i Y_i$ has a mean of $\sum_{i=1}^{d} \mathbb{E}[X_i]\mathbb{E}[Y_i] = 0$ and a variance of $\sum_{i=1}^{d} \text{Var}(X_i Y_i) = d$. By the law of large numbers, $\frac{X \cdot Y}{d} \xrightarrow{\mathbb{P}} 0$ as $d \to \infty$. Similarly, $\frac{\|X\|^2}{d} \xrightarrow{\mathbb{P}} 1$ and $\frac{\|Y\|^2}{d} \xrightarrow{\mathbb{P}} 1$. Thus, their cosine similarity converges to zero:

$$\frac{X \cdot Y}{\|X\|\|Y\|} = \frac{X \cdot Y/d}{\sqrt{(\|X\|^2/d)(\|Y\|^2/d)}} \xrightarrow{\mathbb{P}} 0.$$

Hence, as the dimension $d$ increases, any pair of independent standard Gaussian vectors becomes almost orthogonal.

The hubness phenomenon can then be interpreted via the Pythagorean theorem: the squared distance between two independent samples $x$ and $y$ is approximately the sum of their squared distances to the origin, $\|x - y\|^2 \approx \|x\|^2 + \|y\|^2$. As a result, points slightly closer to the origin than others tend to appear as nearest-neighbors for many points, becoming hubs due to their relative centrality, while points far from the origin become antihubs (see Figure 3a). Importantly, removing those central points just causes the next most central points to become hubs. Hubness is not caused by a few outliers, but an inherent property of the distribution (Radovanovic et al., 2010).

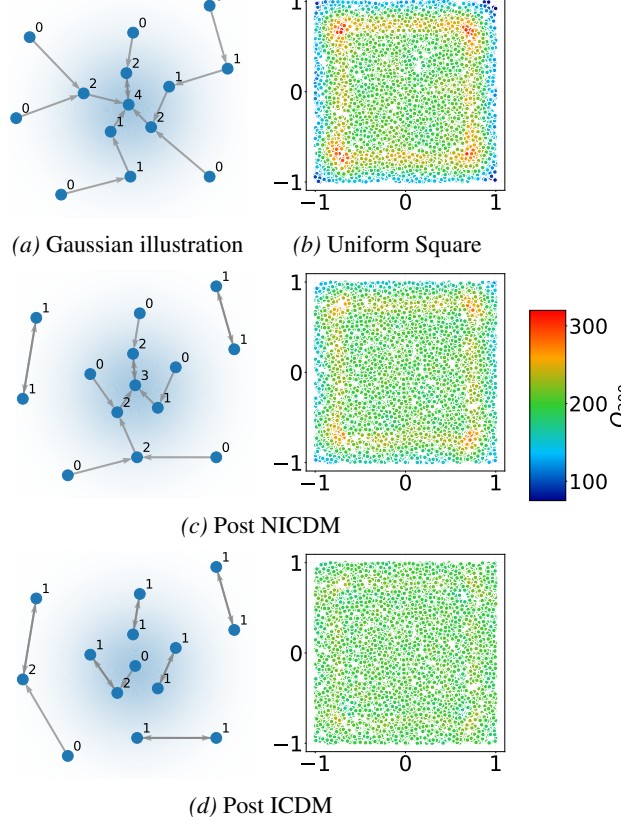

*(a) Gaussian illustration*      *(b) Uniform Square*

*(c) Post NICDM*

*(d) Post ICDM*

*Figure 4.* **Density Gradient**: *(a)* (adapted from Hara et al. (2016), Fig. 2c): Illustrative Gaussian points (blue density) with 1-occurrence counts; arrows indicate nearest-neighbors. Occurrences accumulate in denser regions, as arrows follow the gradient of the density. *(b)*: 5000 samples from a uniform square. Color indicates 200-occurrence count. Boundaries force neighbors inward, creating density gradients and hubness; both effects intensify with increasing dimension. After applying *(c)* NICDM and *(d)* ICDM: Left, arrows show updated neighbors. The maximum $O_1$ decreases from 4 to 3 (NICDM) and 2 (ICDM); the number of antihubs drops from 6 to 5 and 2. Right, the occurrence distribution becomes more uniform after NICDM, and even more so after ICDM.

## 2.2. Origin of Hubness: Density Gradient

Hubness is often described as a manifestation of the curse of dimensionality (Radovanovic et al., 2010), but for Gaussian data, it is closely linked to centrality. If centrality is removed by sampling uniformly on a sphere, hubness disappears entirely, even in high dimensions (see Figure 3b).

Low et al. (2013) argue that hubness is fundamentally caused by *density gradients*. As illustrated in Figure 4a, for a Gaussian distribution, samples tend to have their nearest-neighbors in regions of higher density, effectively following the gradient of the density function. Consequently, points in denser regions accumulate more occurrences.

Boundaries also induce density gradients, since the density drops to zero outside. Figure 4b illustrates how boundaries

affect occurrences for a uniform distribution on a square in $d = 2$: near edges, neighbors can only be found inward. The boundary-to-volume ratio increases exponentially with dimension, amplifying this effect: in high dimensions, almost all points are close to a boundary (Low et al., 2013). A $d$-dimensional sphere is a finite, boundary-less $(d - 1)$-dimensional manifold. Sampling uniformly on it avoids hubness, even in high dimensions, because there is no density gradient. However, non-uniform sampling introduces a density gradient and thus hubness. In a non-Gaussian setting, simply projecting points onto a sphere does not eliminate hubness (see Table 4).

The density gradient is intrinsic to the data distribution, so hubness cannot be eliminated simply by increasing the sample size (Low et al., 2013). Furthermore, hubness correlates with the intrinsic rather than the ambient dimension, meaning that reducing the ambient dimension does not alleviate hubness unless information is lost (Radovanovic et al., 2010; Flexer & Schnitzer, 2015). This confirms that hubness is fundamentally tied to the data distribution.

Importantly, hubness is not just a theoretical concern: it impairs distance-based evaluation (see Figure 1) and we observe it in practice within the very embedding spaces used to evaluate generative models (see Appendix A, Table 4).

## 3. Overview of In-sample Hubness Reduction

Restoring meaningful neighborhoods in high-dimensional spaces has motivated the development of numerous hubness reduction methods. They typically compute secondary dissimilarities between data points to replace original distances for nearest-neighbor search. Their performance has been compared across various scenarios in the literature (Feldbauer & Flexer, 2019; Amblard et al., 2022; Obraczka & Rahm, 2022), and we consider the union of their recommendations here.

Among these, Local DisSimilarity (DSL) (Hara et al., 2016) aims to flatten the density directly, while the others are scaling methods that seek to restore symmetry in nearest-neighbor relationships by scaling distances. By forcing symmetry, these scaling methods consistently increase nearest-neighbor classification accuracy (Feldbauer & Flexer, 2019), indicating that they improve the semantic correctness of local neighbor relationships by reducing the influence of one-way relationships induced by hubness. The Gaussian variant of Mutual Proximity ($MP^{Gauss}$) (Schnitzer et al., 2012) uses all pairwise distances for scaling, whereas the remaining methods are local scaling approaches that use the local neighborhood of each point to scale distances. These include Local Scaling (LS) (Zelnik-Manor & Perona, 2004), Cross-domain Similarity Local Scaling (CSLS) (Lample et al., 2018), and the Non-Iterative Contextual Dissimilar-

ity Measure (NICDM) (Jégou et al., 2007; 2010), whose formulations are closely related.

Given the pairwise distances $\{d_{x_i,x_j}\}_{i,j=1}^N$ between a set of points $\{x_i\}_{i=1}^N$, NICDM produces secondary dissimilarities as follows:

$$\text{NICDM}(d_{x_i,x_j}) = d_{x_i,x_j} \frac{\bar{\mu}}{\sqrt{\mu_i \mu_j}}$$

where

$$\mu_i = \frac{1}{K} \sum_{k=1}^{K} d_{x_i, \text{NN}_k(x_i)},$$

$\text{NN}_k(x)$ is the $k$-th nearest-neighbor of $x$, and $\bar{\mu}$ is the average of all $\mu_i$. Since $\bar{\mu}$ is a global constant, it can be ignored in practice (Schnitzer et al., 2012). The resulting dissimilarity is low when both $\sqrt{\frac{d_{x_i,x_j}}{\mu_i}}$ and $\sqrt{\frac{d_{x_j,x_i}}{\mu_j}}$ are small (i.e., for reciprocal nearest-neighbors). Local scaling methods have $O(N^2)$ complexity.

Many hubness reduction methods were originally proposed in other contexts and only later adapted for hubness reduction. The Contextual Dissimilarity Measure (CDM) was first introduced for image retrieval to improve the symmetry of neighbor relationships (Jégou et al., 2007). Later, only the simpler Non-Iterative version (NICDM) was adapted for hubness reduction, while the *Iterative* version (ICDM) was dismissed as too computationally expensive without yielding significant improvement (Schnitzer et al., 2012). Subsequent works followed this recommendation and used only NICDM (Feldbauer & Flexer, 2019; Amblard et al., 2022; Obraczka & Rahm, 2022). ICDM has not been considered for hubness reduction since then.

We compared these methods for hubness reduction (see Table 5), and found that despite being overlooked, ICDM consistently achieves the best performance across all 17 tested pairs of datasets and embeddings. Figures 4c and 4d illustrate the effects of NICDM and ICDM. Both reduce hubness, as evidenced by the decrease in maximum $O_1$ and the number of antihubs (left), and by the more uniform occurrence distribution (right).

The *Iterative* CDM (ICDM) is obtained by repeatedly applying the non-iterative version to progressively improve the symmetry of neighbor relationships. Let $^{(t)}$ denote quantities at iteration $t$. The update rule is:

$$d_{x_i,x_j}^{(t+1)} = d_{x_i,x_j}^{(t)} \frac{\bar{\mu}^{(t)}}{\sqrt{\mu_i^{(t)} \mu_j^{(t)}}}$$

These iterations minimize the disparity among the $\mu_i^{(t)}$ values, $S^{(t)} = \sum_{i=1}^{N} |\mu_i^{(t)} - \bar{\mu}^{(t)}|$, converging to a fixed point where all $\mu_i^{(t)}$ are equal (Jégou et al., 2010). Defining

$\delta_i = \prod_{t=1}^{T} \sqrt{\frac{\bar{\mu}^{(t)}}{\mu_i^{(t)}}}$, the final result can be expressed using the original distances: $d_{x_i,x_j}^{(T)} = d_{x_i,x_j} \delta_i \delta_j$.

In image retrieval, ICDM is first applied to a base dataset. Then, for a query point $x^q$, its nearest-neighbors in the base dataset are found using:

$$\text{NN}_k(x^q) = \text{k-argmin}_i \, d_{x^q,x_i} \delta_i$$

Finding the nearest base neighbors of $x^q$ does not require knowing the update term $\delta_q$ for the query point, as it is constant across all comparisons (Jégou et al., 2010). However, for out-of-sample evaluation, it is not sufficient to simply rank the neighbors of generated points. For this, *distances* to neighbors are required. ICDM does not provide these directly, so Section 5 presents an algorithmic solution.

To the best of our knowledge, only NICDM, not ICDM, has been considered previously for hubness reduction.

## 4. Related Work: Generative Model Metrics

In this section, we review existing fidelity and coverage metrics for generative model evaluation.

Fidelity metrics usually assess whether generated samples are realistic by measuring their proximity to real samples. Improved Precision (Kynkäänniemi et al., 2019) computes the fraction of generated samples within the $k$-nearest-neighbor radius of real samples: Precision $= \frac{1}{M} \sum_{j=1}^{M} \mathbf{1}_{x_j^g \in \cup_{i=1}^{N} B(x_i^r, \text{NND}_k^r(x_i^r))}$. Density (Naeem et al., 2020) extends this by counting how many real samples each generated sample is close to: Density $= \frac{1}{kM} \sum_{j=1}^{M} \sum_{i=1}^{N} \mathbf{1}_{x_j^g \in B(x_i^r, \text{NND}_k^r(x_i^r))}$.

Conversely, coverage metrics evaluate how well generated samples cover the real data distribution. Improved Recall (Kynkäänniemi et al., 2019) measures the fraction of real samples within the $k$-nearest-neighbor radius of generated samples: Recall $= \frac{1}{N} \sum_{i=1}^{N} \mathbf{1}_{x_i^r \in \cup_{j=1}^{M} B(x_j^g, \text{NND}_k^g(x_j^g))}$. Coverage (Naeem et al., 2020) instead checks if each real sample is covered by at least one generated point within its real ball: Coverage $= \frac{1}{N} \sum_{i=1}^{N} \mathbf{1}_{\exists j, x_j^g \in B(x_i^r, \text{NND}_k^r(x_i^r))}$.

Other metrics use similar constructions. For example, Precision Cover and Recall Cover (Cheema & Urner, 2023) count balls as covered if at least $k' > 1$ points lie within them. P-precision and P-recall (Park & Kim, 2023) use kernel density estimates instead of hard balls. Rather than using the union of balls as an approximate support, $\alpha$-Precision and $\beta$-Recall (Alaa et al., 2022) use a one-class approach to estimate supports at varying levels, while Topological Precision and Recall (Kim et al., 2023) use topologically conditioned density kernels. More closely related, Clipped Density and Clipped Coverage (Salvy et al., 2026) are variants of Density and Coverage that limit the influence of individual points to aggregate scores. In Clipped Density, for example, real ball radii are clipped to reduce outlier effects, and the result is normalized by the real set score.

Hubness directly distorts these distance-based metrics: generated hubs artificially cover many real points, inflating coverage scores, while antihub regions act as blind spots where generated samples receive zero fidelity. This distortion severely biases the evaluation, yielding misleading conclusions about generative model performance.

The work most closely related to ours identified a flaw in distance-based metrics and noted that hubness is "very closely related" to their analysis, though it is not explicitly targeted (Khayatkhoei & AbdAlmageed, 2023). Their focus is on the support constructed by Precision and Recall. For a uniform hypersphere in high dimensions, the support induced by Precision is biased toward the center, leading to different outcomes for points at the same distance, depending on whether they are inwards or outwards from the sphere. To restore symmetry, they propose using two alternative supports: one based on real points and one on generated points, so that when one is biased, the other can be used. symPrecision and symRecall are then defined as the minimum of Precision or Recall and its complement. However, for non-unimodal distributions, both supports can be biased simultaneously. As shown in Figure 1, sym metrics fail for a bimodal hypersphere. In contrast, GICDM addresses hubness more generally and is not tied to a specific metric.

## 5. GICDM

Our goal is to enable reliable distance-based evaluation of generative models in high-dimensional embedding spaces, where hubness distorts nearest-neighbor relationships. To address this, we identify three key desiderata for effective hubness mitigation:

1. **Reduce hubness in the real dataset:** In the transformed distance space, the local density of real data should be uniform, so that nearest-neighbor relationships are meaningful and symmetric.

2. **Preserve the relative positioning of generated points:** Hubness mitigation should maintain the true local geometric structure between generated points and the real data manifold, removing only hubness effects.

3. **Conditional independence of generated points given the real set:** Distance-based metrics give sample-wise scores (e.g., a fidelity score to each individual generated sample). So, the distance scaling of each generated point should depend only on its relationship to the real set in the original space, so that a given generated sample falls into the same real balls, receiving the same fidelity score, regardless of the other generated samples evaluated alongside it.

## 5.1. Density Estimation from Nearest-Neighbors Distances

ICDM iteratively applies a popularity penalty: it expands the space around points in relatively high-density regions (those with small $\mu_i$, which receive large $\delta_i$), making them harder to reach, and contracts the space around points in relatively low-density regions (i.e. large $\mu_i$, small $\delta_i$), making them easier to reach. This process repeats until the average neighbor distance is nearly equal for all points.

**Proposition 5.1.** *Let*

$$\hat{p}_{\mu,K}(x_i) \stackrel{\text{def}}{=} \frac{1}{NV_d} \frac{1}{\mu_i^d} \left( \frac{1}{K} \sum_{k=1}^{K} k^{1/d} \right)^d$$

*where $V_d$ is the volume of the unit ball in dimension $d$. Then, $\hat{p}_{\mu,K}(x_i)$ is a local density estimator.*

The proof is provided in Appendix B.1. $\hat{p}_{\mu,K}$ is closely related to the $K$-NN density estimator (Loftsgaarden & Quesenberry, 1965), but uses the average $K$-NN distance $\mu_i$ instead of the $K$-th nearest-neighbor distance.

**Corollary 5.2.** *At ICDM convergence, when $\forall i, |\mu_i - \bar{\mu}| < \epsilon$ for an arbitrarily small $\epsilon > 0$, the local density estimates $\hat{p}_{\mu,K}(x_i)$ are equal for all $x_i$.*

Thus, ICDM effectively uniformizes the data density, removing density gradients and thereby reducing hubness.

Applying ICDM directly to the union of real and generated points violates all of our desiderata: the uniformized density would be that of the combined set, not just the real data (violating desideratum 1). Secondary dissimilarities would be influenced by generated points, making each generated point's evaluation depend on the others (violating desideratum 3), and dissimilarities between real points would be affected by generated data (violating all desiderata).

## 5.2. Extending ICDM to Generated Points

Let $\{x_i^r\}_{i=1}^N$ denote real data points and $\{x_j^g\}_{j=1}^M$ denote generated data points. Superscripts $^r$ and $^g$ indicate quantities associated with real and generated points, respectively.

To uniformize the density of the real points only (desideratum 1), we first apply ICDM to the real set, yielding scaling factors $\{\delta_i^r\}_{i=1}^N$ such that secondary dissimilarities between real points are $\text{ICDM}(x_i^r, x_j^r) = d_{x_i^r, x_j^r} \delta_i^r \delta_j^r$.

We now define secondary dissimilarities between generated and real points as $d'_{x_i^r, x_j^g} = d_{x_i^r, x_j^g} \delta_i^r \delta_j^g$, where $\delta_j^g$ is to be determined for each generated point $x_j^g$. To satisfy desiderata 2 and 3, $\delta_j^g$ must be computed independently for each generated point, using only its relationship to the real set.

For real points, considering $K$ neighbors, the average neighbor distance is $\mu^r(x_i) = \frac{1}{K} \sum_{k=1}^{K} d_{x_i, \text{NN}_k(x_i)} =$

$\frac{1}{K} \sum_{m=1}^{K+1} D_m^{r,i}$, where $D_m^{r,i}$ is the $m$-th smallest distance from $x_i$ to the real set, with $D_1^{r,i} = 0$ (distance to itself). Thus, for generated points, the appropriate neighborhood depth in terms of order statistics is $K + 1$, not $K$.

Let $^{eq}$ denote quantities at equilibrium after applying ICDM to the real set. For all $i$, $|\mu_i^{r,eq} - \overline{\mu^{r,eq}}| < \epsilon$ for an arbitrarily small $\epsilon > 0$, i.e., the average neighbor distance is practically equalized across real points.

Assuming that a generated sample $x_j^g$ is drawn from the same distribution as the real samples, we require it to exhibit the same spatial properties as the real data. In particular, its average neighbor distance must match that of the real samples: $|\mu_j^{g,eq} - \overline{\mu^{r,eq}}| < \epsilon$. By enforcing exact equality at the theoretical equilibrium $\mu_j^{g,eq} = \overline{\mu^{r,eq}}$, we can write for all $i, j$,

$$\mu_j^{g,eq} = \frac{1}{K+1} \sum_{k=1}^{K+1} \delta_j^g \delta_{\text{NN}_k^{r,eq}(x_j^g)}^r d_{x_j^g, \text{NN}_k^{r,eq}(x_j^g)}$$

$$= \delta_j^g \frac{1}{K+1} \sum_{k=1}^{K+1} \delta_{\text{NN}_k^{r,eq}(x_j^g)}^r d_{x_j^g, \text{NN}_k^{r,eq}(x_j^g)},$$

where $\text{NN}_k^{r,eq}$ denotes the $k$-th nearest real neighbor after ICDM (known using $\delta_i^r$). This yields:

$$\delta_j^g = \frac{\overline{\mu^{r,eq}}}{\frac{1}{K+1} \sum_{k=1}^{K+1} \delta_{\text{NN}_k^{r,eq}(x_j^g)}^r d_{x_j^g, \text{NN}_k^{r,eq}(x_j^g)}} \tag{1}$$

Thus, we define the unfiltered secondary dissimilarity as $\text{GICDM}_{\text{unfiltered}}(x_i^r, x_j^g) = d_{x_i^r, x_j^g} \delta_i^r \delta_j^g$, with $\delta_j^g$ as above.

This assumes that generated points are drawn from the same distribution as the real data. In practice, however, we do not know a priori whether a generated point comes from the real data distribution, this is precisely what we aim to assess.

## 5.3. Preventing Overcorrections

If a generated point is not well aligned with the real data manifold and we are outside the crossover regime (discussed in the next subsection), its distances to its real neighbors, $d_{x_j^g, \text{NN}_k^{r,eq}(x_j^g)}$, will not be consistent with those observed among real points. Consequently, the scaling factor $\delta_j^g$ computed for this generated point will differ substantially from the $\delta_i^r$ values of its real neighbors.

The $\delta$ values quantify the scaling applied to each point's neighborhood. A significant discrepancy between $\delta_j^g$ and those of its real neighbors indicates that the generated point does not lie within the same region as the real data.

To address this, we compare $\delta_j^g$ to the average $\delta$ of its real neighbors, $\bar{\delta}_j^{r|g} = \frac{1}{K+1} \sum_{k=1}^{K+1} \delta_{\text{NN}_k^{r,eq}(x_j^g)}^r$, and filter out generated points for which $\left| \bar{\delta}_j^{r|g} - \delta_j^g \right| / \bar{\delta}_j^{r|g}$ exceeds the $q$-

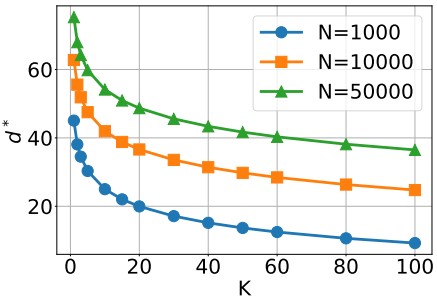

Figure 5. **Crossover dimension** $d^*$: At $d^*$ for a standard Gaussian, the median $K$-NN squared distance and the expected squared distance to the center equalize. Points are closer to each other below $d^*$ and closer to the center above. The plot shows $d^*$ vs $K$ using Proposition 5.3, for various sample sizes $N$. As $N$ increases, points become closer so $d^*$ increases; as $K$ increases, $K$-NN distances increase so $d^*$ decreases. Using two sufficiently different $K$ values ensures at least one avoids the crossover regime.

quantile of $\{|\bar{\delta}_i^{r|r} - \delta_i^r|/\bar{\delta}_i^{r|r}\}_{i=1}^N$ computed over real points, where $\bar{\delta}_i^{r|r} = \frac{1}{K}\sum_{k=1}^K \delta_{NN_k^{r,eq}(x_i^r)}^r$ and $q$ is set to $0.95$ in our experiments. These filtered points fall outside the real data manifold. As they are not in any real ball, their fidelity score is zero and they do not contribute to coverage.

### 5.4. Multi-Scale Filtering

A generated point not well aligned with the real data manifold can have similar distances to its real neighbors as those observed among real points, thus erroneously avoiding the filtering, if the local density structure falls into the *crossover regime*, which depends on the density, $N$, $d$ and $K$.

To illustrate this, consider samples from a standard Gaussian. In low dimensions, points are closer to each other than to the center, while the reverse occurs in high dimensions. In between is a transition regime where points are as far from each other as from the center. In this intermediate regime, a generated point can be "far" from the real data (e.g., in the empty center), yet still have a $\delta$ value similar to its real neighbors, because its distances to them are similar to their distances to their own neighbors.

**Proposition 5.3.** *Let $X_1, \ldots, X_N$ be i.i.d. samples from $\mathcal{N}(0, I_d)$. The* crossover dimension $d^*$ *such that the median squared $k$-nearest-neighbor distance equals the expected squared distance to the center,* $\mathrm{median}(NND_k(X_i)^2) = \mathbb{E}\|X_i\|^2$, *is the solution of*

$$\int_0^\infty \mathbb{P}\left(\mathrm{Bin}(N-1, F_{\chi^2_{d^*}(\lambda=r)}(d^*)) \geq k\right) f_{\chi^2_{d^*}}(r) dr = \frac{1}{2}$$

*where $f_{\chi^2_{d^*}}(r)$ is the density of the $\chi^2_{d^*}$ distribution and $F_{\chi^2_{d^*}(\lambda)}$ is the cumulative distribution function of the noncentral $\chi^2_{d^*}$ distribution with noncentrality parameter $\lambda$.*

The proof is in Appendix B.2. Using Proposition 5.3, we

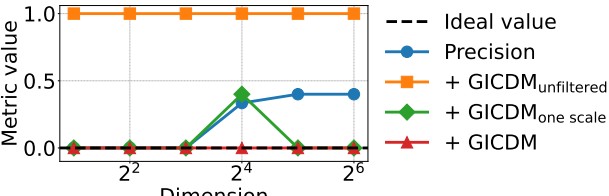

Figure 6. **Hypersphere Test ablation**: In the hypersphere scenario from Figure 1, GICDM components are incrementally added to Precision. Unfiltered GICDM overcorrects generated points. Single-scale filtering reduces this effect, but a spike remains when 5-NN distances between real points and to generated points are similar. Finally, multi-scale filtering achieves the ideal result.

numerically estimate $d^*$. Figure 5 plots $d^*$ versus $K$ for various $N$. As expected, increasing $N$ brings points closer, raising $d^*$, while increasing $K$ increases $K$-NN distances, lowering $d^*$. Crucially, $d^*$ depends on both $N$ and $K$.

While this analysis relies on the Gaussian case to quantify the crossover regime, the underlying mechanism applies generally. For any data distribution, the $K$-th nearest-neighbor distance monotonically increases with $K$. Consequently, changing $K$ inherently shifts the scale of local neighborhoods and, by extension, shifts the targeted crossover regime where out-of-manifold distances coincidentally match local in-manifold distances. By choosing two sufficiently different $K$ values for filtering (e.g., $K_2 = 10K_1$), we can ensure that at least one is outside the problematic crossover regime.

In summary, our proposed Generative Iterative Contextual Dissimilarity Measure (GICDM) combines the three discussed components to address our desiderata: (1) applying ICDM to the real set to uniformize its density (desideratum 1), (2) computing scaling factors for generated points based solely on their relationship to the real set (desiderata 2 and 3), and (3) filtering generated points at multiple scales to prevent overcorrection (desideratum 2). The complete algorithm is summarized in Algorithm 1.

Figure 6 demonstrates the effect of each GICDM component in the hypersphere test from Figure 1, tracking the Precision metric as components of GICDM are introduced (out-of-sample ICDM: GICDM$_{\text{unfiltered}}$, filtering and multi-scale filtering). The results indicate that all three components are required to fully resolve hubness-related issues.

GICDM is a dissimilarity rather than a distance metric as it does not obey the triangle inequality. However, this is not an issue for distance-based generative model evaluation, which only requires a measure of similarity capable of ranking neighbors and determining neighborhood membership, two operations that do not rely on the triangle inequality. This aligns with the common use of other non-metric dissimilarities, such as cosine similarity, in feature spaces (Oquab et al., 2023; Siméoni et al., 2025).

**Algorithm 1** GICDM

**Require:** Real points $\{x_i^r\}_{i=1}^N$, generated points $\{x_j^g\}_{j=1}^M$, neighborhood sizes $K_1, K_2$, quantile $q$

1: **for** $K \in \{K_1, K_2\}$ **do**
2:    Apply ICDM to $\{x_i^r\}_{i=1}^N$ with neighborhood size $K$ to obtain scaling factors $\{\delta_{i,K}^r\}_{i=1}^N$
3:    **for** $i = 1$ to $N$ **do**
4:       $\bar{\delta}_{i,K}^{r|r} \leftarrow \frac{1}{K} \sum_{k=1}^K \delta_{\mathrm{NN}_k^{r,eq}(x_i^r),K}^r$
5:       $r_{i,K}^r \leftarrow \frac{|\bar{\delta}_{i,K}^{r|r} - \delta_{i,K}^r|}{\bar{\delta}_{i,K}^{r|r}}$
6:    $T_K \leftarrow q$-th quantile of $\{r_{i,K}^r\}_{i=1}^N$
7: **for** $j = 1$ to $M$ **do**
8:    $keep \leftarrow$ True
9:    **for** $K \in \{K_1, K_2\}$ **do**
10:       Compute $\delta_{j,K}^g$ using Eq. (1) with $\{\delta_{i,K}^r\}$
11:       $\bar{\delta}_{j,K}^{r|g} \leftarrow \frac{1}{K+1} \sum_{k=1}^{K+1} \delta_{\mathrm{NN}_k^{r,eq}(x_j^g),K}^r$
12:       $r_{j,K}^g \leftarrow \frac{|\bar{\delta}_{j,K}^{r|g} - \delta_{j,K}^g|}{\bar{\delta}_{j,K}^{r|g}}$
13:       **if** $r_{j,K}^g > T_K$ **then**
14:          $keep \leftarrow$ False
15:    **for** $i = 1$ to $N$ **do**
16:       **if** $keep$ **then**
17:          $\mathrm{GICDM}(x_i^r, x_j^g) \leftarrow d(x_i^r, x_j^g) \cdot \delta_{i,K_1}^r \cdot \delta_{j,K_1}^g$

# 6. Experiments

All experiments were conducted on a single NVIDIA H100 GPU with 80GB of memory and use 10 iterations for ICDM (see Appendix C for a convergence analysis).

For metrics that use a neighborhood size $k$, we set the GICDM neighborhood size to $2k$. This ensures the $k$-th nearest-neighbor distance remains comparable across points after hubness correction. While ICDM equalizes the average distance to the $K$-nearest-neighbors ($\mu_i^{r,eq}$), the $K$-th distance itself can still vary, especially for small $K$. Using $K = 2k$ places the $k$-th neighbor near the center of the averaged distances, improving stability.

**Synthetic Benchmark.** We evaluated standard metrics and their GICDM-corrected versions on the extensive synthetic benchmark proposed by Räisä et al. (2025). This benchmark evaluates metric scores evolution across 14 *Purpose* scenarios, whether metrics yield the expected values (e.g., 1 for realistic generated data) in 13 *Bounds* cases, and 3 *Other* miscellaneous tests. Testing over 10 metrics (details in Appendix D), we found that GICDM consistently improves two Purpose tests. Table 1 summarizes the results for the best-performing fidelity and coverage metrics: GICDM substantially improves performance, passing 23 criteria instead of 18 for Clipped Density, and 22 instead of 18 for Clipped Coverage. This underlines that hubness indeed causes several metric failures and that GICDM successfully corrects

*Table 1.* **Benchmark success count.** Passed tests count (higher is better) on the synthetic benchmark of Räisä et al. (2025), before and after applying GICDM. Test categories: *Purpose* (does the score evolution match the intended test objective), *Bounds* (do metrics yield the exact expected values, e.g., 1 in the ideal case), and *Other* (additional checks). Results are shown for the best-performing fidelity and coverage metrics: Clipped Density and Clipped Coverage, as well as for Precision and Recall.

|  | PURPOSE | BOUNDS | OTHER |
|---|---|---|---|
| PRECISION | 6/14 | 3/13 | **2**/3 |
| CLIPPED DENSITY | 8/14 | 8/13 | **2**/3 |
| **+ GICDM (OURS)** | **10**/14 | **11**/13 | **2**/3 |
| RECALL | 5/14 | 2/13 | 0/3 |
| CLIPPED COVERAGE | 8/14 | 9/13 | **1**/3 |
| **+ GICDM (OURS)** | **10**/14 | **11**/13 | **1**/3 |

*Table 2.* **Correlation with human scores.** Pearson correlation between metric scores and human error rates. Results are not significant for FFHQ or other embeddings. GICDM maintains or improves Clipped Density's (the best-performing metric) correlation with human scores, especially for DINOv3, which exhibits more hubness.

| EMBED. | METRIC | CIFAR 10 | IMAGE NET | LSUN BEDROOM |
|---|---|---|---|---|
|  | PRECISION | $-0.82$ | 0.76 | - |
| DINOv2 | CLIPPED DENSITY | 0.93 | 0.75 | 0.81 |
|  | + GICDM | 0.97 | 0.80 | 0.80 |
|  | PRECISION | 0.86 | 0.74 | 0.96 |
| DINOv3 | CLIPPED DENSITY | 0.82 | 0.67 | 0.94 |
|  | + GICDM | 0.95 | 0.82 | 0.97 |

them without altering already-satisfied criteria.

**Real Data and Human Correlation.** We further evaluated 42 datasets generated by various models on CIFAR-10 (Krizhevsky et al., 2009), ImageNet (Deng et al., 2009), LSUN Bedroom (Yu et al., 2015), and FFHQ (Kazemi & Sullivan, 2014), using the data shared by Stein et al. (2023). We then computed the Pearson correlation between fidelity metrics and human error rates (where volunteers discriminated real from generated images). As shown in Table 2, GICDM consistently maintains or improves the correlation of Clipped Density (the best-performing fidelity metric) with human judgments, particularly for DINOv3 embeddings that are heavily affected by hubness. This alignment confirms that GICDM preserves the relative positioning of generated samples (desideratum 2) while mitigating hubness. Since desiderata 1 and 3 are satisfied by construction, GICDM fulfills all desiderata. See Appendix E.3 for further details; notably, as also demonstrated by an additional benchmark on real CIFAR10 data (Appendix F), GICDM is only effective when metrics are robust to real outliers, further highlighting the importance of using robust metrics.

**Classifier-Free Guidance.** Table 3 shows the effect of

*Table 3.* **Effect of guidance.** Metric scores for DINOv2 embeddings of ImageNet-trained DiT-XL-2, with and without classifier-free guidance. Guidance is expected to increase fidelity and decrease coverage. Clipped Density increases both with and without GICDM, but Clipped Coverage decreases only with GICDM.

| METRIC | DiT-XL-2 | + GUIDANCE |
|---|---|---|
| CLIPPED DENSITY | 0.74 | 1.62 ↗ |
| + GICDM | 0.86 | 1.15 ↗ |
| CLIPPED COVERAGE | 0.64 | 0.83 ↗ |
| + GICDM | 0.79 | 0.55 ↘ |

classifier-free guidance on DiT-XL-2 for ImageNet with DINOv2 embeddings. Both prior best metrics, Clipped Density and Clipped Coverage, increase with guidance, which is counter-intuitive since guidance should improve fidelity but reduce coverage. With GICDM, Clipped Density increases, but Clipped Coverage decreases as expected. These results highlight how hubness can distort metrics, and how GICDM corrects it.

**Qualitative Visualization.** Figure 15 qualitatively shows GICDM's effect on LSUN Bedroom with DINOv2 embeddings: before applying GICDM, the most frequent generated hub appears in 64 real neighborhoods, while after applying GICDM, it appears in only 4. These 4 images are more semantically similar to the hub than the average 64 were. Thus, GICDM effectively mitigates hubness and restores a more semantically meaningful neighborhood structure.

In summary, by removing hubness, GICDM consistently improves metric results, confirming that hubness caused the observed evaluation issues and that GICDM resolves them without removing desirable data structure. This leads to more trustworthy evaluation of generative models and better alignment with human judgment.

## 7. Conclusion

In this work, we introduced GICDM, a hubness mitigation method specifically designed for generative model evaluation. GICDM leverages ICDM to uniformize the density of the real data manifold, and computes out-of-sample scaling factors for generated samples based solely on their relationship to the real set. To prevent overcorrection, a multi-scale filtering strategy discards points with inconsistent local scaling.

Extensive experiments on both synthetic and real benchmarks demonstrate that GICDM resolves failures of existing metrics, restores reliable fidelity and coverage scores, and improves alignment with human judgment. While applicable to any distance-based metric, a limitation is that its effectiveness relies on the underlying metric being robust to real outliers, such as Clipped Density and Clipped Coverage.

Overall, GICDM enables more reliable and trustworthy evaluation of generative models in high dimensional spaces.

Code to reproduce the experiments and use GICDM is available at https://github.com/nicolassalvy/GICDM.

## Acknowledgements

This work benefited from state aid managed by the Agence Nationale de la Recherche under the France 2030 programme, reference ANR-22-PESN-0012 and from the European Union's Horizon 2020 Research Infrastructures Grant EBRAIN-Health 101058516. This work was performed using HPC resources from GENCI-IDRIS (Grant 2025-AD011014887R2).

## Impact Statement

This paper presents work whose goal is to advance the field of Generative Model Evaluation. By improving the assessment of generative models, our work can help guide the development of more reliable models, which is important for data-scarce domains such as healthcare. At the same time, we recognize that generative models may be misused (e.g., for deepfakes).

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

# A. Hubness Measures

Hubness skews the distribution of $k$-occurrences, $O_k$. To quantify hubness, Low et al. (2013) proposed examining the right tail of the distribution by measuring the largest $k$-occurrence values and averaging them for robustness:

$$h_1^k(q) = \frac{1}{k|\mathcal{O}_k(q)|} \sum_{x \in \mathcal{O}_k(q)} O_k(x),$$

where $\mathcal{O}_k(q)$ is the set of the $\lfloor qn \rfloor$ points with the highest $k$-occurrences, and typically $q = 0.01$ is used for a small proportion. $h_1^k(q)$ measures, on average, how much more often the top $q$-fraction of points appear among $k$-nearest-neighbors compared to the average point.

On the opposite side of the distribution, Flexer & Schnitzer (2015) proposed measuring the proportion of antihubs, $A^k$, defined as points that never appear in the $k$-nearest-neighbors of any other point:

$$A^k = \frac{|\{x \in D \mid O_k(x) = 0\}|}{|D|}.$$

Figure 7 shows that both measures indicate increasing hubness for standard Gaussian data as dimensionality grows.

### A.1. Hubness in Common Embedding Spaces

Table 4 reports results for Cifar10 (Krizhevsky et al., 2009), FFHQ (Kazemi & Sullivan, 2014), ImageNet (Deng et al., 2009), and LSUN Bedroom (Yu et al., 2015) datasets embedded with VGG16 (Simonyan & Zisserman, 2015), Inceptionv3 (Szegedy et al., 2016), DINOv2 (Oquab et al., 2023), and DINOv3 (Siméoni et al., 2025) image encoders, as well as the Free Music Archive (Defferrard et al., 2017) dataset embedded with the CLAP audio encoder (Elizalde et al., 2023). All exhibit hubness. DINOv2, which has the lowest dimensionality (1024) among the image encoders (VGG16: 4096, Inceptionv3: 2048, DINOv3: 4096), is the least affected, but still shows hubness.

For the least affected case, ImageNet with DINOv2 embeddings, the top 1% most frequent points appear more than 4 times as often as the average point among the 5-nearest-neighbors of other points, and 10% of points are antihubs, never appearing in any 5-nearest-neighbors list. This hubness affects the reliability of nearest-neighbor relationships, which are crucial for distance-based generative model evaluation metrics.

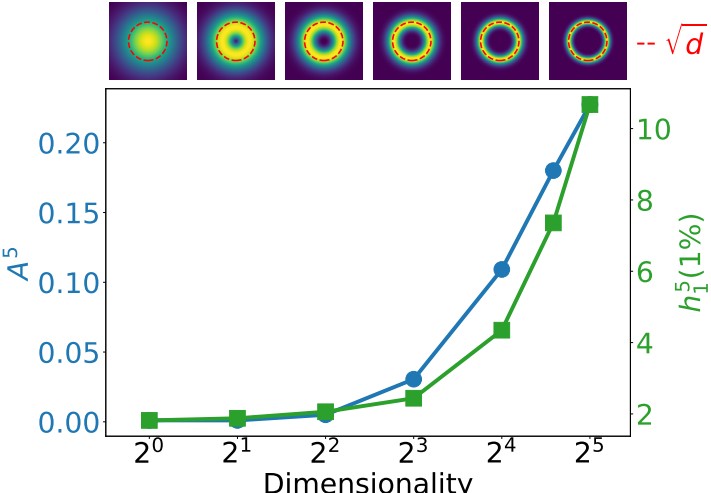

*Figure 7.* **Hubness Evolution** for standard Gaussian data ($N = 20000$) as dimensionality increases. *Top*: Intuitive 2D visualization of a Gaussian using a wrapped chi-squared distribution with $d$ degrees of freedom (lighter colors indicate higher density). As the dimension increases, samples concentrate on a thin spherical shell. *Bottom*: The y-axes are not shared. Both hubness measures ($A^5$ and $h_1^5(1\%)$) indicate that hubness increases with dimensionality.

*Table 4.* **Hubness measures for various datasets and embedding spaces**. $A^5$ is the proportion of antihubs ($O_5(x) = 0$), and $h_1^5(1\%)$ quantifies how much more frequently the top 1% of points appear among $k$-nearest-neighbors compared to the average. In the absence of hubness, the $k$-occurrence distribution would be nearly symmetric around its mean (as in Figure 2a), yielding $h_1^5(1\%)$ values slightly above 2 and $A^5$ values below 0.01. All reported values are at least twice as large, indicating pronounced hubness.

| EMBEDDING | DIMENSION | DATASET | $h_1^5(1\%)$ | $A^5$ |
|---|---|---|---|---|
| VGG16 | 4096 | CIFAR10 | 10.4 | 0.21 |
| | | FFHQ | 9.8 | 0.23 |
| | | IMAGENET | 12.6 | 0.20 |
| | | LSUN BEDROOM | 16.6 | 0.31 |
| INCEPTIONV3 | 2048 | CIFAR10 | 9.0 | 0.22 |
| | | FFHQ | 8.3 | 0.20 |
| | | IMAGENET | 5.7 | 0.20 |
| | | LSUN BEDROOM | 12.9 | 0.28 |
| DINOV2 | 1024 | CIFAR10 | 6.5 | 0.14 |
| | | FFHQ | 5.8 | 0.12 |
| | | IMAGENET | 4.4 | 0.10 |
| | | LSUN BEDROOM | 5.8 | 0.13 |
| DINOV3 | 4096 | CIFAR10 | 9.0 | 0.24 |
| | | FFHQ | 10.1 | 0.26 |
| | | IMAGENET | 8.0 | 0.20 |
| | | LSUN BEDROOM | 9.0 | 0.23 |
| CLAP | 1024 | FREE MUSIC ARCHIVE | 4.9 | 0.09 |

## A.2. Hubness Reduction Comparison

Table 5 compares various hubness reduction methods on the datasets and embeddings from Table 4, showing that ICDM constently outperforms other methods. See Appendix J for full results.

*Table 5.* **Hubness reduction comparison.** For each method, we report its rank and value for $A^5$ and $h_1^5(1\%)$, averaged over five datasets and embeddings from Table 4. "Sphere" is projection onto the unit sphere. Lower values are better. ICDM is best in all 17 cases. For methods needing a neighborhood size $K$, we took the $K$ in $\{5, 10, 20, 50\}$ with lowest $h_1^5(1\%)$. With $h_1^5(1\%)$ below 2 and $A^5$ near 0, ICDM eliminates hubness effectively.

| METHOD | $h_1^5(1\%)$ | | $A^5$ | |
|---|---|---|---|---|
| | RANK | VALUE | RANK | VALUE |
| ORIGINAL | 7.8 | 8.8 | 7.9 | 0.20 |
| SPHERE | 6.9 | 6.0 | 7.1 | 0.12 |
| MP$^{\text{GAUSS}}$ | 5.7 | 4.5 | 5.9 | 0.08 |
| LS | 5.2 | 4.1 | 5.1 | 0.05 |
| CSLS | 3.7 | 3.2 | 4.1 | 0.04 |
| DSL | 3.2 | 3.3 | 2.0 | 0.01 |
| NICDM | 2.6 | 3.1 | 3.0 | 0.03 |
| ICDM | 1.0 | 1.7 | 1.0 | 0.00 |

# B. Proofs

## B.1. Proof of Proposition 5.1

**Proposition 5.1.** *Let*

$$\hat{p}_{\mu,K}(x_i) \stackrel{\text{def}}{=} \frac{1}{NV_d} \frac{1}{\mu_i^d} \left( \frac{1}{K} \sum_{k=1}^{K} k^{1/d} \right)^d$$

*where $V_d$ is the volume of the unit ball in dimension d. Then, $\hat{p}_{\mu,K}(x_i)$ is a local density estimator.*

*Proof.* Let $\hat{p}_{\text{K-NN}}$ denote the $K$-nearest-neighbors density estimator, by definition (Loftsgaarden & Quesenberry, 1965):

$$\hat{p}_{\text{K-NN}}(x_i) = \frac{K}{NV_d(\text{NND}_K(x_i))^d}$$

where $\text{NND}_K(x_i)$ is the distance from $x_i$ to its $K$-th nearest-neighbor, $N$ is the number of samples, and $V_d$ is the volume of the unit ball in $d$ dimensions. We have:

$$\text{NND}_k(x_i) = \left( \frac{k}{NV_d\hat{p}_{\text{K-NN}}(x_i)} \right)^{1/d}$$

$$\mu_i = \frac{1}{K} \sum_{k=1}^{K} \left( \frac{k}{NV_d\hat{p}_{\text{k-NN}}(x_i)} \right)^{1/d}$$

$$\mu_i = \frac{1}{(NV_d)^{1/d}} \frac{1}{K} \sum_{k=1}^{K} k^{1/d} \hat{p}_{\text{k-NN}}(x_i)^{-1/d}$$

Substituting this expression for $\mu_i$ into the definition of $\hat{p}_{\mu,K}(x_i)$ yields:

$$\hat{p}_{\mu,K}(x_i) = \frac{1}{NV_d} \left( \frac{1}{(NV_d)^{1/d}} \frac{1}{K} \sum_{k=1}^{K} k^{1/d} \hat{p}_{\text{k-NN}}(x_i)^{-1/d} \right)^{-d} \left( \frac{1}{K} \sum_{k=1}^{K} k^{1/d} \right)^d$$

$$= \left( \frac{\sum_{k=1}^{K} k^{1/d}}{\sum_{k=1}^{K} k^{1/d} \hat{p}_{\text{k-NN}}(x_i)^{-1/d}} \right)^d$$

$$= \left( \sum_{k=1}^{K} \underbrace{\frac{k^{1/d}}{\sum_{j=1}^{K} j^{1/d}}}_{w_{K,k}} \hat{p}_{\text{k-NN}}(x_i)^{-1/d} \right)^{-d}$$

The weights $w_{K,k}$ satisfy the conditions of a regular summability method (Toeplitz, 1911; Tucciarone, 1973):

- $w_{K,k} \xrightarrow{K \to \infty} 0$ for any fixed $k \in \mathbb{N}$,

- are positive and $\sum_{k=1}^{K} w_{K,k} = 1$, so $\sum_{k=1}^{K} |w_{K,k}| = \sum_{k=1}^{K} w_{K,k} = 1$, and the absolute sum of the weights is bounded by a constant independent of $K$.

- $\sum_{k=1}^{K} w_{K,k} = 1$, so $\sum_{k=1}^{K} w_{K,k} \xrightarrow{K \to \infty} 1$.

With $k(N)$ such that $k(N) \xrightarrow{N \to \infty} \infty$, and $k(N)/N \xrightarrow{N \to \infty} 0$, $\hat{p}_{\text{k-NN}}(x_i) \xrightarrow{P} p(x_i)$.

By the Silverman-Toeplitz theorem (Toeplitz, 1911; Tucciarone, 1973), which guarantees that regular summability methods preserve limits, we have $\sum_{k=1}^{K} \frac{k^{1/d}}{\sum_{j=1}^{K} j^{1/d}} \hat{p}_{\text{k-NN}}(x_i)^{-1/d} \xrightarrow{P} p(x_i)^{-1/d}$ as $K \to \infty$ and $K/N \to 0$.

Finally, because the power function is continuous, $\hat{p}_{\mu,K}(x_i) \xrightarrow{P} p(x_i)$. □

## B.2. Proof of Proposition 5.3

**Proposition 5.3.** *Let $X_1, \ldots, X_N$ be i.i.d. samples from $\mathcal{N}(0, I_d)$. The* crossover dimension $d^*$ *such that the median squared $k$-nearest-neighbor distance equals the expected squared distance to the center,* $\mathrm{median}(NND_k(X_i)^2) = \mathbb{E}\|X_i\|^2$, *is the solution of*

$$\int_0^\infty \mathbb{P}\left(\mathrm{Bin}(N-1, F_{\chi^2_{d^*}(\lambda=r)}(d^*)) \geq k\right) f_{\chi^2_{d^*}}(r) dr = \frac{1}{2}$$

*where $f_{\chi^2_{d^*}}(r)$ is the density of the $\chi^2_{d^*}$ distribution and $F_{\chi^2_{d^*}(\lambda)}$ is the cumulative distribution function of the noncentral $\chi^2_{d^*}$ distribution with noncentrality parameter $\lambda$.*

*Proof.* Let $X_1, \ldots, X_N$ be i.i.d. samples from $\mathcal{N}(0, I_d)$. The squared distance to the center satisfies $\|X_1\|^2 \sim \chi^2_d$, so $\mathbb{E}\|X_1\|^2 = d$.

For a fixed $X_1$, $\forall j = 2, \ldots, N$, $X_j - X_1 \mid X_1 \sim \mathcal{N}(-X_1, I_d)$. Conditionally on $\|X_1\|^2 = r$, the random variables $\{\|X_j - X_1\|^2\}_{j=2}^N$ are i.i.d., and we have $\|X_j - X_1\|^2 \mid \|X_1\|^2 = r \sim \chi^2_d(\lambda = r)$, a noncentral chi-square distribution with $d$ degrees of freedom and noncentrality parameter $\lambda$, with cumulative distribution function $F_{\chi^2_d(\lambda=r)}$.

$\mathrm{NND}_k(X_1)^2$ is the $k$-th order statistic of these $N-1$ variables. Its c.d.f. is such that:

$$\mathbb{P}\left(\mathrm{NND}_k(X_1)^2 \leq t \mid \|X_1\|^2 = r\right) = \mathbb{P}\left(\mathrm{Bin}(N-1, F_{\chi^2_d(\lambda=r)}(t)) \geq k\right)$$

(see e.g. Mood (1950) section VI 5.1).

So, after integration to remove the conditioning on $\|X_1\|^2 = r$, we have:

$$\mathbb{P}\left(\mathrm{NND}_k(X_1)^2 \leq t\right) = \int_0^\infty \mathbb{P}\left(\mathrm{Bin}(N-1, F_{\chi^2_d(\lambda=r)}(t)) \geq k\right) f_{\chi^2_d}(r) dr$$

By definition, the median of $\mathrm{NND}_k(X_1)^2$ is the value $t$ for which $\Pr\left(\mathrm{NND}_k(X_1)^2 \leq t\right) = \frac{1}{2}$. When $t = d$, this gives the equation in the proposition. $\square$

## C. Empirical Convergence of ICDM

GICDM relies on ICDM uniformizing the density of the real data manifold, i.e., $|\mu_i^r - \bar{\mu}^r| < \epsilon$ for all $i$ for a small $\epsilon$ after a finite number of iterations. In this section, we empirically validate this pointwise convergence across various datasets and embeddings.

We used 16 combinations of embedders (DINOv2, DINOv3, Inceptionv3, VGG16) and datasets (CIFAR-10, ImageNet, FFHQ, LSUN-bedroom) and monitored the relative difference between individual distance averages $\mu_i^r$ and their overall mean $\bar{\mu}^r$, using $K = 10$ and $K = 100$ (the two values used when evaluating metrics with $k = 5$).

The results, shown in Figure 8, demonstrate fast pointwise convergence. Across all tested scenarios, the relative deviation values $\frac{|\mu_i^r - \bar{\mu}^r|}{\bar{\mu}^r}$ fall within the following ranges:

- $[0.84, 1.51]$ after 1 iteration (equivalent to NICDM, the non-iterative version),

- $[0.985, 1.022]$ after 5 iterations,

- $[0.9989, 1.0017]$ after 10 iterations (the stopping criterion used in our experiments),

- $[0.99983, 1.00018]$ after 15 iterations,

- $[0.999966, 1.000036]$ after 20 iterations.

After 10 iterations, the maximum deviation of any individual $\mu_i^r$ from the mean is less than $0.17\%$ across all 16 diverse scenarios. Consequently, the pointwise convergence assumption holds robustly in practice, and stopping at 10 ensures convergence for GICDM.

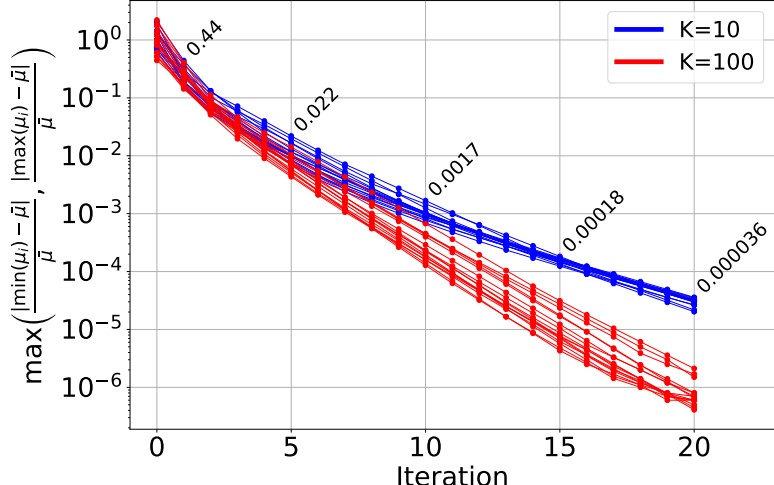

*Figure 8.* **Empirical convergence of ICDM.** Maximum relative deviation of individual neighborhood distance averages $\mu_i^r$ from their overall mean $\bar{\mu}^r$ across iterations, for 16 combinations of datasets and embeddings, with $K = 10$ and $K = 100$ (the two values used when evaluating metrics with $k = 5$). Values from the above results at iterations 1, 5, 10, 15 and 20 show the biggest relative difference. After 10 iterations, the maximum deviation is less than $0.17\%$ across all scenarios, confirming fast pointwise convergence.

# D. Synthetic Benchmark

## D.1. Setup

[Räisä et al. (2025)](#) recently proposed a synthetic benchmark to evaluate fidelity and coverage metrics for generative models, from which they concluded that no existing fidelity or coverage metric was fully satisfactory. In this section, we evaluate the impact of GICDM on this benchmark.

We apply several changes to the benchmark to tackle issues pointed out recently ([Salvy et al., 2026](#)):

- We used `export PYTHONHASHSEED=42` to ensure reproducibility across runs.

- For tests with constant sample size, we used 10000 samples instead of 1000 to remove instability issues. To mitigate partially the increased computational cost, we used 10 or 11 values per tests instead of 20 or 51. For Precision Cover

*Table 6.* **Fidelity metrics benchmark with and without GICDM**. Tests are grouped by desiderata type: *Purpose* checks whether the curve shape of the evaluated metric matches the intended test objective, while *Bounds* evaluates whether the metric yields the expected values (e.g., 1 for perfect generated data). For each metric, T indicates a passed test, and F indicates failure. Adding GICDM improves Purpose results on several tests, such as "Gaussian Std. Deviation Difference" and "Hypersphere Surface", where the real data is fixed and the generated data varies in radius (variance for a Gaussian or radius for a sphere). GICDM improves Bounds results for the normalized metric Clipped Density.

| DESIDERATA | SANITY CHECK | α-PRECISION | PRECISION COVER | PRECISION | +GICDM | DENSITY | +GICDM | SYMPRECISION | +GICDM | P-PRECISION | +GICDM | CLIPPED DENSITY | +GICDM |
|---|---|---|---|---|---|---|---|---|---|---|---|---|---|
| | DISCRETE NUM. VS. CONTINUOUS NUM. | F | F | F | F | F | F | F | F | F | F | F | F |
| | GAUSSIAN MEAN DIFFERENCE | T | T | T | T | T | T | T | T | T | T | T | T |
| | GAUSSIAN MEAN DIFFERENCE + OUTLIER | T | T | F | T | T | T | F | T | T | T | T | T |
| | GAUSSIAN MEAN DIFFERENCE + PARETO | T | T | T | T | T | T | T | T | T | T | T | T |
| | GAUSSIAN STD. DEVIATION DIFFERENCE | F | F | F | T | F | T | F | F | F | T | F | T |
| | HYPERCUBE, VARYING SAMPLE SIZE | F | F | F | F | F | F | F | F | F | F | F | F |
| PURPOSE | HYPERCUBE, VARYING SYN. SIZE | F | F | F | F | F | F | F | F | F | F | F | F |
| | HYPERSPHERE SURFACE | T | F | F | T | F | T | T | T | F | T | F | T |
| | MODE COLLAPSE | T | F | F | T | T | T | T | T | T | T | T | T |
| | MODE DROPPING + INVENTION | F | F | T | T | T | T | F | F | T | T | T | T |
| | ONE DISJOINT DIM. + MANY IDENTICAL DIM. | F | F | F | F | F | F | F | F | F | F | F | F |
| | SEQUENTIAL MODE DROPPING | F | F | T | T | T | T | F | F | T | T | T | T |
| | SIMULTANEOUS MODE DROPPING | F | F | T | T | T | T | F | F | T | T | T | T |
| | SPHERE VS. TORUS | T | F | T | T | T | T | T | T | T | T | T | T |
| HYPERPARAM. | HYPERCUBE, VARYING SYN. SIZE | T | F | T | T | T | T | F | F | T | T | T | T |
| DATA | HYPERCUBE, VARYING SAMPLE SIZE | F | F | F | F | F | F | F | F | F | F | F | F |
| | DISCRETE NUM. VS. CONTINUOUS NUM. | F | F | F | F | F | F | F | F | F | F | F | F |
| | GAUSSIAN MEAN DIFFERENCE | F | T | F | F | T | F | F | F | F | F | T | T |
| | GAUSSIAN MEAN DIFFERENCE + OUTLIER | F | T | F | F | F | F | F | F | F | F | F | T |
| | GAUSSIAN MEAN DIFFERENCE + PARETO | T | T | F | F | T | F | T | F | T | F | T | T |
| | GAUSSIAN STD. DEVIATION DIFFERENCE | F | F | F | F | F | F | F | F | F | F | F | T |
| | HYPERSPHERE SURFACE | F | F | F | F | F | F | T | F | F | F | F | T |
| BOUNDS | MODE COLLAPSE | F | T | F | F | T | F | F | F | F | F | T | T |
| | MODE DROPPING + INVENTION | F | F | T | F | T | F | F | F | F | F | T | T |
| | ONE DISJOINT DIM. + MANY IDENTICAL DIM. | F | F | F | F | F | F | F | F | F | F | F | F |
| | SCALING ONE DIMENSION | T | T | T | T | T | T | T | T | T | T | T | T |
| | SEQUENTIAL MODE DROPPING | F | F | F | F | T | F | F | F | F | F | T | T |
| | SIMULTANEOUS MODE DROPPING | F | F | F | F | T | F | F | F | F | F | T | T |
| | SPHERE VS. TORUS | F | T | T | T | T | T | T | T | T | T | T | T |
| INVARIANCE | SCALING ONE DIMENSION | T | T | T | T | T | T | T | T | T | T | T | T |

*Table 7.* **Coverage metrics benchmark with and without GICDM**. This table, analogous to Table 6, reports results for coverage metrics. T indicates a passed test, while F indicates failure. For coverage metrics, L and H denote that the metric correctly identifies low (L) or high (H) coverage scenarios; either outcome is considered a pass. As with fidelity metrics, GICDM improves Purpose results on several tests, such as "Gaussian Std. Deviation Difference" and "Hypersphere Surface." GICDM also improves Bounds results for the normalized metric Clipped Coverage.

| Desiderata | Sanity Check | β-Recall | Recall Cover | Recall | + GICDM | Coverage | + GICDM | SymRecall | + GICDM | P-recall | + GICDM | Clipped Coverage | + GICDM |
|---|---|---|---|---|---|---|---|---|---|---|---|---|---|
| Purpose | Discrete Num. vs. Continuous Num. | F | F | F | F | F | F | F | F | F | F | F | F |
| | Gaussian Mean Difference | T | T | T | T | T | T | T | T | T | T | T | T |
| | Gaussian Mean Difference + Outlier | T | T | F | T | T | T | T | T | T | T | T | T |
| | Gaussian Mean Difference + Pareto | T | T | T | T | T | T | T | T | T | T | T | T |
| | Gaussian Std. Deviation Difference | L | F | F | H | F | L | L | L | F | H | F | L |
| | Hypercube, Varying Sample Size | F | F | F | F | F | F | F | F | F | F | F | F |
| | Hypercube, Varying Syn. Size | F | F | F | F | F | F | F | F | F | F | F | F |
| | Hypersphere Surface | T | F | F | T | F | T | T | T | F | T | F | T |
| | Mode Collapse | F | L | F | F | F | F | F | F | F | F | L | L |
| | Mode Dropping + Invention | F | F | H | H | F | F | F | H | H | H | L | L |
| | One Disjoint Dim. + Many Identical Dim. | F | F | F | F | F | F | F | F | F | F | F | F |
| | Sequential Mode Dropping | F | T | T | T | T | T | T | T | T | T | T | T |
| | Simultaneous Mode Dropping | F | T | T | T | T | T | T | T | T | T | T | T |
| | Sphere vs. Torus | T | F | F | T | F | T | F | T | F | T | T | T |
| Hyperparam. | Hypercube, Varying Syn. Size | F | F | F | F | F | F | F | F | F | F | F | F |
| Data | Hypercube, Varying Sample Size | F | F | F | F | F | F | F | F | F | F | F | F |
| Bounds | Discrete Num. vs. Continuous Num. | F | F | F | F | F | F | F | F | F | F | F | F |
| | Gaussian Mean Difference | F | T | F | F | T | F | F | F | F | F | T | T |
| | Gaussian Mean Difference + Outlier | F | T | F | F | T | F | F | F | F | F | T | T |
| | Gaussian Mean Difference + Pareto | F | T | F | F | T | F | T | F | T | F | T | T |
| | Gaussian Std. Deviation Difference | F | F | F | F | F | F | T | F | F | F | F | L |
| | Hypersphere Surface | F | F | F | F | F | F | T | F | F | F | F | F |
| | Mode Collapse | F | T | F | F | T | F | F | F | F | F | T | T |
| | Mode Dropping + Invention | F | T | T | F | T | F | T | F | F | F | T | T |
| | One Disjoint Dim. + Many Identical Dim. | T | F | F | F | F | F | F | F | F | F | F | F |
| | Scaling One Dimension | T | T | F | T | T | T | T | T | T | T | T | T |
| | Sequential Mode Dropping | F | T | F | F | T | F | F | F | F | F | T | T |
| | Simultaneous Mode Dropping | F | T | F | F | T | F | F | F | F | F | T | T |
| | Sphere vs. Torus | T | F | T | T | F | F | F | T | T | T | T | T |
| Invariance | Scaling One Dimension | T | T | F | T | T | T | T | T | T | T | T | T |

and Recall Cover, we used the implementation from Salvy et al. (2026) for faster computation.

- We corrected the success criterion for coverage metrics in the "Mode Dropping + Invention" test: "L" instead of failure for a decrease in coverage as invented modes are added.

We made one additional change to the success criteria, specifically for the "Gaussian Std. Deviation Difference" test. In this test, the real data is sampled from a standard Gaussian distribution, while the generated data is sampled from a Gaussian with the same mean but varying standard deviation. The test is conducted in dimensions 1, 8, and 64.

In the original benchmark, the success criterion for fidelity metrics was to observe a high fidelity score for low standard deviation values (up to 1), followed by a decrease in fidelity as the standard deviation of the generated data increased beyond 1. This is intuitive in low dimensions, where a generated point from a distribution with lower variance than the real data will fall into denser regions of the real data distribution, resulting in high fidelity.

However, in high dimensions, Gaussian distributions resemble spheres with an empty center. As a result, a generated point from a Gaussian with lower standard deviation than the real data will be located in the empty center region, far from any real points, and should therefore have low fidelity. Consequently, we modified the success criterion for this test in dimension $64$ to match that of the "Hypersphere Surface" test, which is the analogous test using spheres instead of Gaussians. For simplicity, we removed the criterion for dimension $8$, as the ideal behavior in this intermediate dimension is not clear.

### D.2. Results

We ran the benchmark on standard metrics as well as on distance-based metrics with GICDM correction. Results are shown in Tables 6 and 7. T indicates a passed test, while F indicates failure. For coverage metrics, the benchmark distinguishes between high (H, support-based) diversity metrics and low (L, density-based) coverage metrics. Either is considered a success as long as the metric consistently exhibits one behavior: L or H.

GICDM consistently improves Purpose results (shape of the curve) on both the "Gaussian Std. Deviation Difference" and "Hypersphere Surface" tests.

The hypersphere test was already passed by symmetric metrics without GICDM, as these metrics were specifically designed for this scenario. However, symPrecision fails both with and without GICDM on the Gaussian std test, as it exhibits the behavior expected for spheres in both high and low dimensions (i.e., low fidelity for low std values, even in dimension 1). The symmetric approach does not work as soon as there is more than one mode, as shown in Figure 1.

For the Bounds criteria, which evaluate whether metrics yield the exact expected values, we observe clear improvements for Clipped metrics. These metrics are normalized, either relative to the score of the real set or via a closed-form formula, and GICDM preserves this normalization effectively. In contrast, other metrics exhibit less robust normalization after GICDM correction, sometimes resulting in ideal values slightly below 1 (e.g., around 0.9). This suggests that normalization by the real set score, as implemented in Clipped Density, could also be advantageous for other metrics after applying GICDM.

### D.3. Remaining Failures

Even for the top metrics with GICDM, some failures remain. We discuss these below.

*Discrete Num. vs. Continuous Num.*: This test aims to detect when generated data is discrete rather than continuous, or vice versa. None of the metrics is designed to detect this failure mode.

*One Disjoint Dim. + Many Identical Dim.*: The real data is sampled from a standard Gaussian distribution in $d$ dimensions, while the generated data is sampled from a Gaussian that matches the real one in $d-1$ dimensions but has variance 6 times larger in one dimension. As noted by Räisä et al. (2025), for metrics based on Euclidean distances, this difference is averaged out as $d$ increases.

*Hypercube tests: Varying Sample Size and Varying Syn. Size*: In this test from Cheema & Urner (2023), the real and synthetic distributions are uniform on $d$-dimensional hypercubes of side $1$, with overlapping volume $0.2$, and the sample sizes vary. To maintain a fixed overlapping volume of $0.2$, the distance $h$ between the corners of the two hypercubes must satisfy $(1-h)^d = 0.2$, i.e., $h = 1 - 0.2^{1/d}$. Thus, as observed by Räisä et al. (2025), the distance between points in the real and synthetic hypercubes decreases as the dimension increases.

These failures arise from limitations of Euclidean distances. The selection of embedding space and distance metric is critical to representing data in a way that captures its underlying structure. For example, if variations along one dimension are semantically more important than others, using Euclidean distance without adjustment may yield suboptimal results.

# E. Evaluation on Real Datasets

## E.1. Data and Setup

We evaluated metrics on generated datasets publicly released by Stein et al. (2023), using 50000 real and 50000 generated samples for each evaluation, balanced across classes when applicable.

For CIFAR-10, generated data includes samples from: LSGM-ODE (Vahdat et al., 2021), PFGM++ (PFGMPP) (Xu et al., 2023), iDDPM-DDIM (Nichol & Dhariwal, 2021), StudioGAN models (Kang et al., 2023a) (ACGAN-Mod (Odena et al., 2017), BigGAN (Brock et al., 2019), LOGAN (Wu et al., 2019), MHGAN (Turner et al., 2019), ReACGAN (Kang et al., 2021), WGAN-GP (Gulrajani et al., 2017)), StyleGAN-XL (Sauer et al., 2022), StyleGAN2-ada (Karras et al., 2020), RESFLOW (Chen et al., 2019), and NVAE (Vahdat & Kautz, 2020).

For ImageNet, $256 \times 256$ rescaled images were provided by Stein et al. (2023), generated by: ADM (Dhariwal & Nichol, 2021), ADMG (Dhariwal & Nichol, 2021), ADMG-ADMU (Dhariwal & Nichol, 2021), BigGAN (Brock et al., 2019),

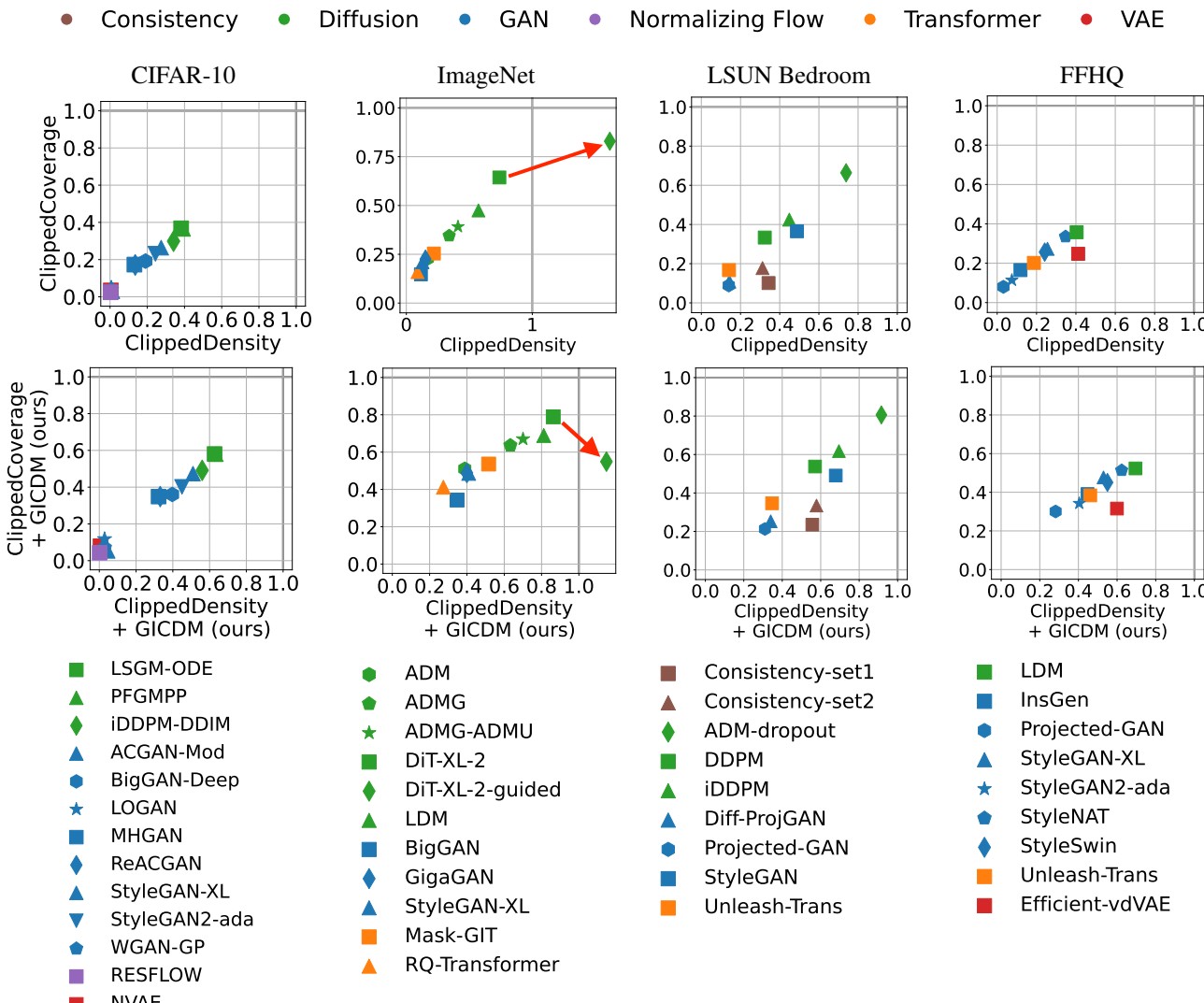

*Figure 9.* **Fidelity vs Coverage in DINOv2 embeddings**: Clipped Density vs Clipped Coverage (top row) and Clipped Density + GICDM vs Clipped Coverage + GICDM (bottom row) for CIFAR-10, ImageNet, LSUN Bedroom, and FFHQ datasets. Each point represents a generative model. For DiT-XL-2 on ImageNet, in the top row, classifier-free guidance increases both fidelity and coverage, whereas with GICDM (bottom row), it increases fidelity but decreases coverage, as expected (red arrows).

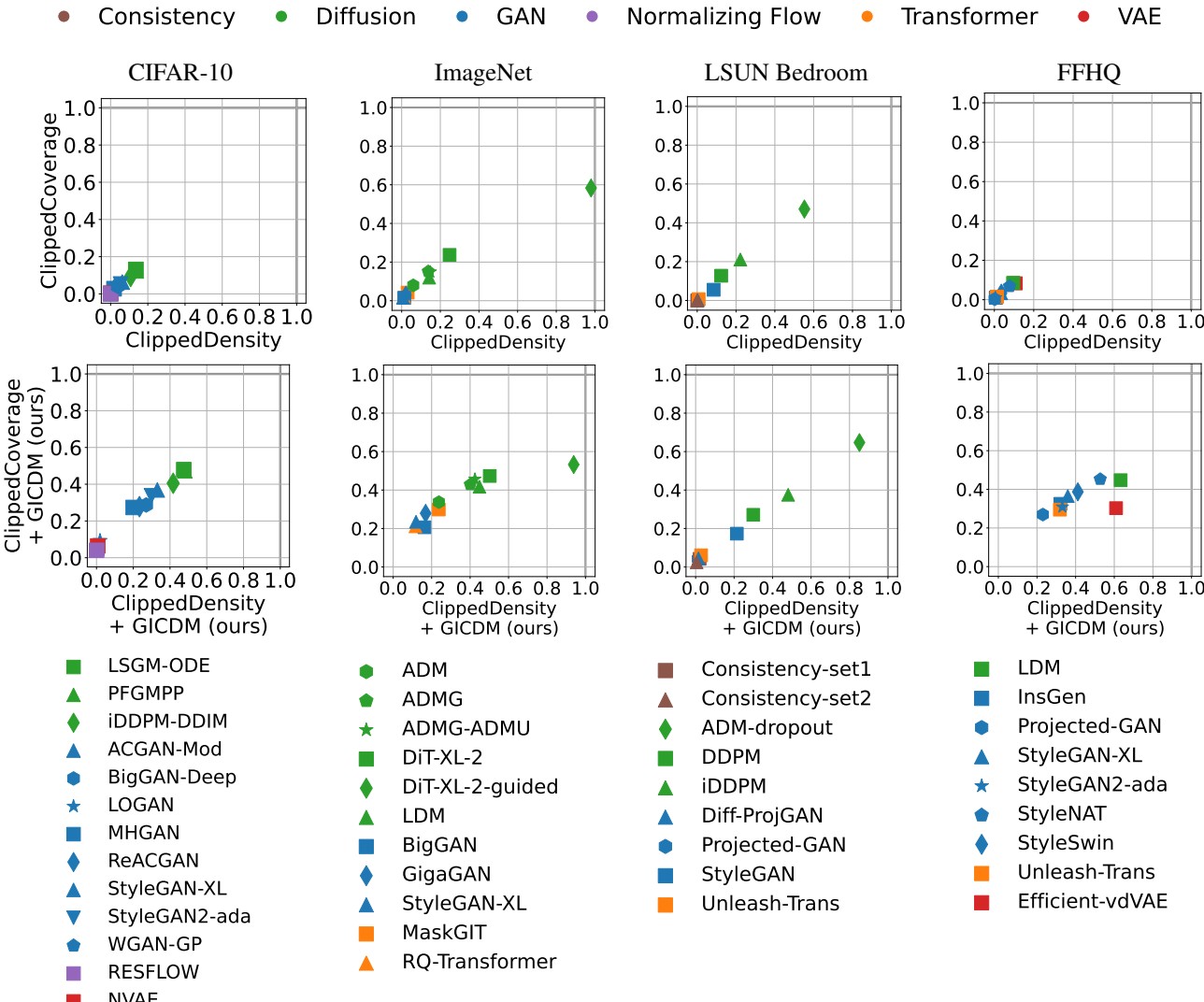

*Figure 10.* **Fidelity vs Coverage in DINOv3 embeddings**: Same setup as Figure 9, but using DINOv3 embeddings. GICDM tends to increase the spread of points.

DiT-XL-2 (Peebles & Xie, 2023), DiT-XL-2-guided (Peebles & Xie, 2023), LDM (Rombach et al., 2022), GigaGAN (Kang et al., 2023b), StyleGAN-XL (Sauer et al., 2022), Mask-GIT (Chang et al., 2022), and RQ-Transformer (Lee et al., 2022).

For LSUN Bedroom (Yu et al., 2015), generated data includes: ADM-dropout (Dhariwal & Nichol, 2021), DDPM (Ho et al., 2020), iDDPM (Nichol & Dhariwal, 2021), StyleGAN (Karras et al., 2019), Diffusion-Projected GAN (Wang et al., 2023), Projected GAN (Sauer et al., 2021), Unleashing Transformers (Bond-Taylor et al., 2022), and two Consistency sets (Song et al., 2023).

For FFHQ (Kazemi & Sullivan, 2014), $256 \times 256$ downsampled images were provided by Stein et al. (2023), generated by: LDM (Rombach et al., 2022), InsGen (Yang et al., 2021), Projected-GAN (Sauer et al., 2021), StyleGAN-XL (Sauer et al., 2022), StyleGAN2-ada (Karras et al., 2020), StyleNAT (Walton et al., 2025), StyleSwin (Zhang et al., 2022), Unleashing Transformers (Bond-Taylor et al., 2022), and Efficient-vdVAE (Hazami et al., 2022).

For consistency with prior work, we set $k = 5$ for all metrics, with default parameters elsewhere.

## E.2. Results

Figures 9 and 10 present fidelity versus coverage plots using Clipped Density and Clipped Coverage, both with and without GICDM, for DINOv2 and DINOv3 embeddings, respectively.

In both embeddings, applying GICDM tends to increase the spread of points. This effect is especially pronounced in DINOv3 embeddings, which are more affected by hubness.

For ImageNet with DINOv2 embeddings, adding guidance to DiT-XL-2 increases both fidelity and coverage in the Clipped measures without GICDM, which is counter-intuitive. In contrast, with GICDM, adding guidance increases fidelity but decreases coverage, as expected.

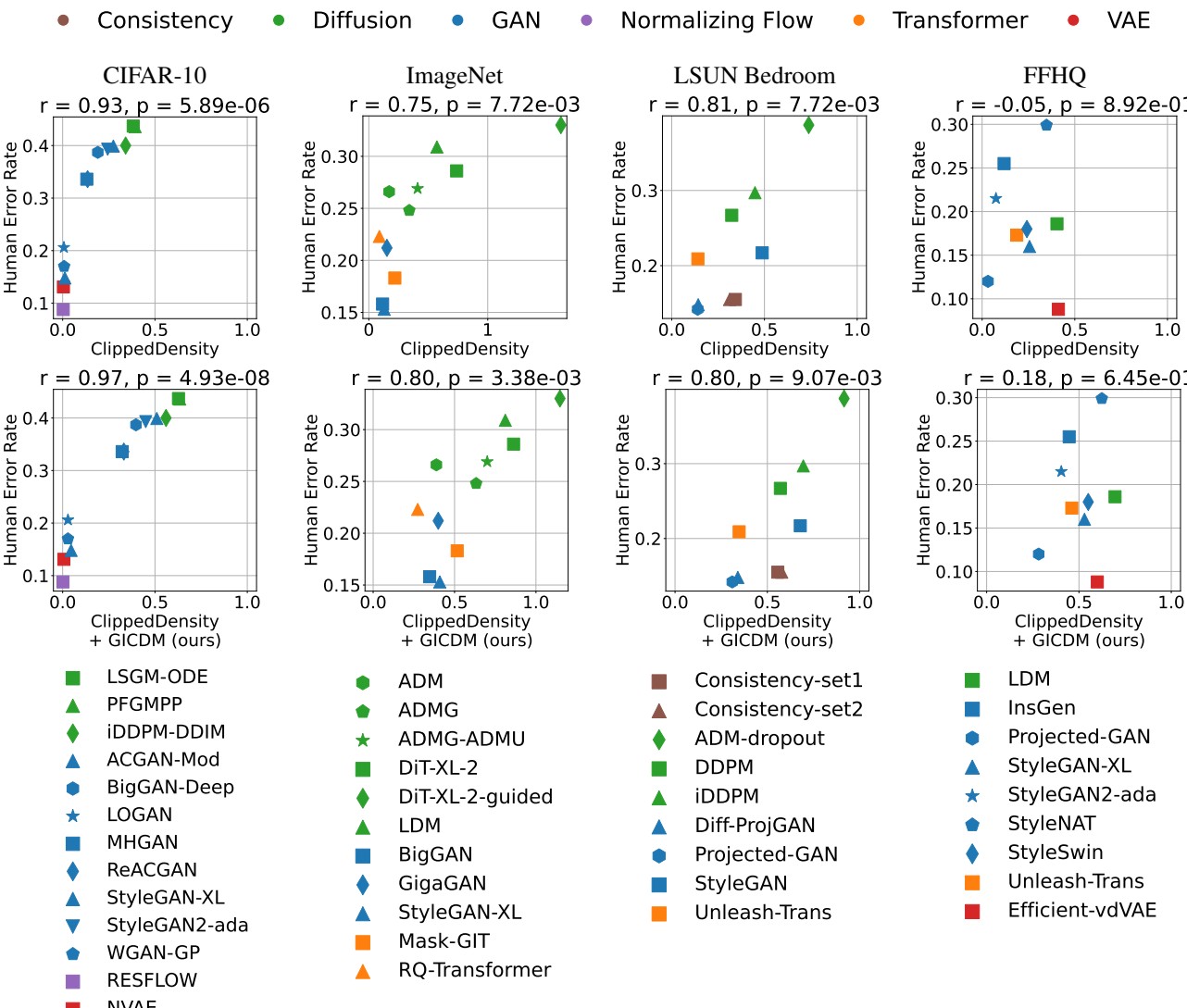

*Figure 11.* **Fidelity vs Human error rates in DINOv2 embeddings**: Human were tasked with discriminating real from generated images. Top row: Clipped Density vs human error rates. Bottom row: Clipped Density + GICDM vs human error rates. Each plot reports the Pearson correlation coefficient $r$ and its $p$-value. Results are not significant for FFHQ, consistent with what was reported for DINOv2 previously (Stein et al., 2023). For other datasets, correlation remains stable or improves slightly with GICDM.

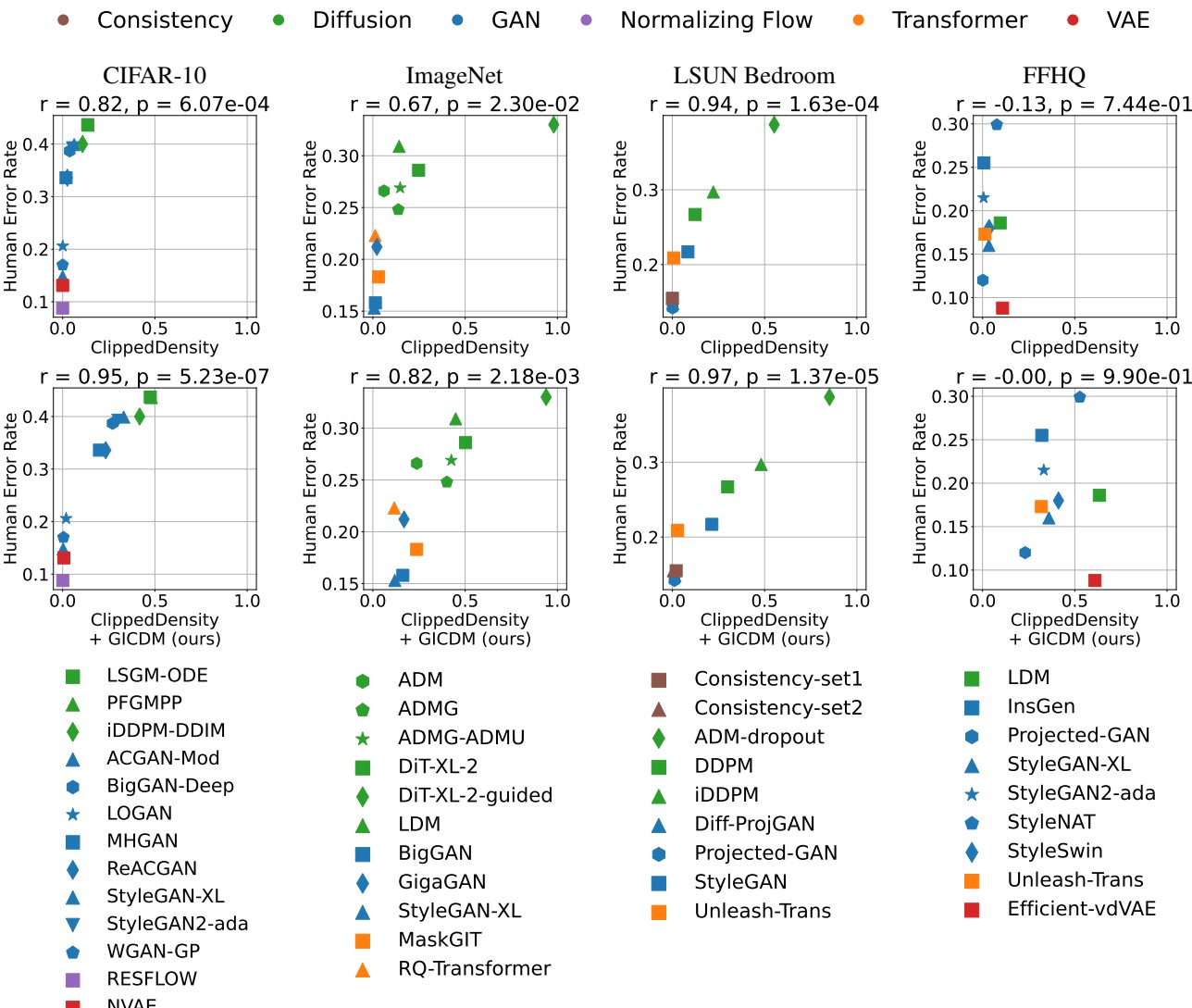

*Figure 12.* **Fidelity vs Human error rates in DINOv3 embeddings**: Same setup as Figure 11, but using DINOv3 embeddings. As with DINOv2, results are not significant for FFHQ. For other datasets, correlation increases with GICDM, and the improvement is more pronounced than with DINOv2, since DINOv3 embeddings are more affected by hubness.

### E.3. Correlation with human scores

Alongside generated images, Stein et al. (2023) also released human error rates for classifying real versus generated images. This discriminator error rate serves as a measure of fidelity: lower error rates indicate higher fidelity of the generated images. We evaluate the correlation between human scores and fidelity metrics, both with and without GICDM.

Figures 11 and 12 show fidelity versus human error rates for DINOv2 and DINOv3 embeddings, respectively, using Clipped Density with and without GICDM. Each plot reports the Pearson correlation coefficient $r$ and its $p$-value. Table 8 summarizes Pearson correlation coefficients for all distance-based metrics, with and without GICDM, on DINOv2 and DINOv3 embeddings. Results for FFHQ are omitted due to lack of significant correlation, as are VGG16 and Inceptionv3 embeddings, which showed almost no significant correlation with any metric.

For Clipped Density and symPrecision, adding GICDM consistently maintains or improves correlation with human scores. For other metrics, this improvement is not observed, and adding GICDM often worsens correlation.

We hypothesize that this lack of improvement is due to insufficient robustness to real outliers, as Precision, Density, and

*Table 8.* **Correlation with human scores.** Significant Pearson correlation coefficients between metric scores and human ratings for each dataset and embedding; non-significant values are indicated by –. The FFHQ column is omitted due to the absence of significant correlations. This is consistent with prior findings using DINOv2 embeddings (Stein et al., 2023). For metrics robust to real outliers (Clipped Density and symPrecision, which equals cPrecision in this context), adding GICDM maintains or improves correlation. For other metrics, adding GICDM generally worsens results, indicating that robustness to real outliers is necessary for GICDM to be effective.

| EMBEDDING | METRIC | CIFAR10 | IMAGENET | LSUN BEDROOM |
|---|---|---|---|---|
| DINOv2 | PRECISION | −0.82 | 0.76 | – |
| | + GICDM | 0.82 | – | −0.78 |
| | DENSITY | −0.91 | 0.68 | – |
| | + GICDM | 0.93 | – | −0.72 |
| | P-PRECISION | 0.98 | 0.75 | 0.69 |
| | + GICDM | 0.82 | – | −0.78 |
| | SYMPRECISION | 0.87 | 0.61 | 0.99 |
| | + GICDM | 0.97 | 0.76 | 0.98 |
| | CLIPPED DENSITY (UNCLIPPED RADII) | 0.73 | −0.85 | – |
| | + GICDM | – | 0.89 | −0.78 |
| | CLIPPED DENSITY | 0.93 | 0.75 | 0.81 |
| | + GICDM | 0.97 | 0.80 | 0.80 |
| DINOv3 | PRECISION | 0.86 | 0.74 | 0.96 |
| | + GICDM | −0.96 | – | – |
| | DENSITY | 0.83 | 0.66 | 0.95 |
| | + GICDM | −0.60 | −0.73 | – |
| | P-PRECISION | 0.97 | 0.79 | 0.97 |
| | + GICDM | −0.96 | – | – |
| | SYMPRECISION | 0.86 | 0.80 | 0.94 |
| | + GICDM | 0.97 | 0.82 | 0.95 |
| | CLIPPED DENSITY (UNCLIPPED RADII) | 0.70 | 0.85 | 0.95 |
| | + GICDM | −0.62 | −0.94 | – |
| | CLIPPED DENSITY | 0.82 | 0.67 | 0.94 |
| | + GICDM | 0.95 | 0.82 | 0.97 |

P-precision are not robust to real outliers (Salvy et al., 2026). To test this, we computed Clipped Density without radius clipping, thereby removing its robustness to real outliers. In this case, it exhibited the same lack or worsening of correlation as the other non-robust metrics (Table 8).

Therefore, robustness to real outliers appears to be a necessary condition for GICDM to improve correlation with human scores, further supporting the recommendation to avoid non-robust metrics.

# F. Real data Benchmark

Salvy et al. (2026) introduced a benchmark for fidelity and coverage metrics, primarily using DINOv2 embeddings of real CIFAR10 data. In this section, we evaluate distance-based metrics with GICDM on this benchmark. We omit their "translating synthetic Gaussian test" on synthetic data, as it is similar to the "Gaussian Mean Difference + Outlier" test in Appendix D.

Compared to the results reported by Salvy et al. (2026) without GICDM, two main differences arise. First, with GICDM, P-precision and P-recall now succeed in the "Introducing bad real and synthetic samples" test. Second, when progressively replacing good synthetic samples with bad ones for fidelity, only Clipped Density and symPrecision succeed, whereas

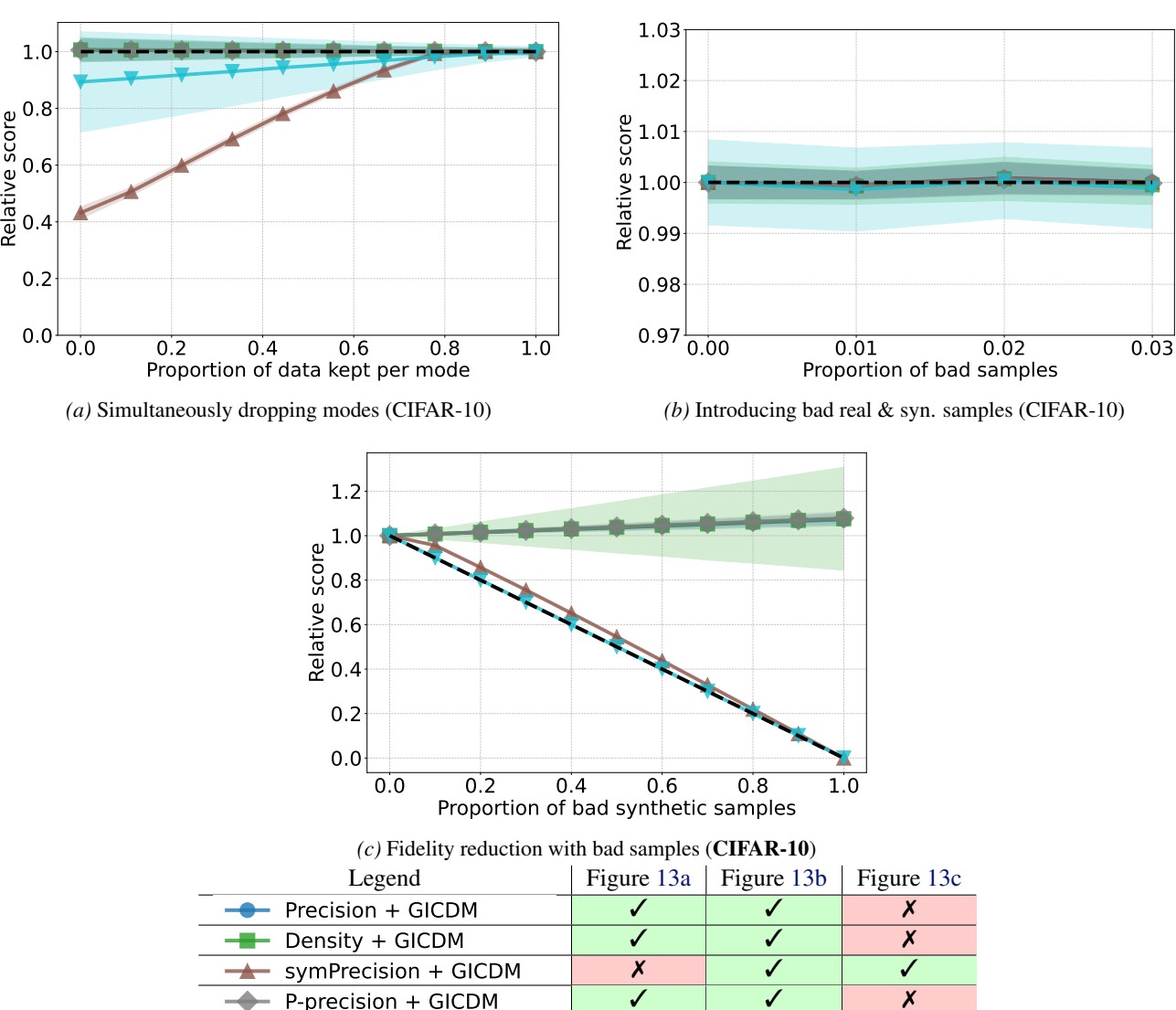

*(a)* Simultaneously dropping modes (CIFAR-10)

*(b)* Introducing bad real & syn. samples (CIFAR-10)

*(c)* Fidelity reduction with bad samples (**CIFAR-10**)

| Legend | | Figure 13a | Figure 13b | Figure 13c |
|---|---|:---:|:---:|:---:|
| ● | Precision + GICDM | ✓ | ✓ | ✗ |
| ■ | Density + GICDM | ✓ | ✓ | ✗ |
| ▲ | symPrecision + GICDM | ✗ | ✓ | ✓ |
| ◆ | P-precision + GICDM | ✓ | ✓ | ✗ |
| ▼ | ClippedDensity + GICDM | ✓ | ✓ | ✓ |
| – – · | Ideal | | | |

*(d)* Legend and summary

*Figure 13.* **Real data tests for fidelity metrics**. Fidelity metrics with GICDM are evaluated in several scenarios. (a) Simultaneous mode dropping: synthetic data from all but one CIFAR-10 class is progressively replaced with data from the remaining class. This test is related to "Dropping + Invention" in Table 6, but on real data. Similarly, only symPrecision fails. (b) Introducing real and synthetic out-of-distribution samples at equal rates: this tests the stability of the metrics. All succeed. (c) Progressively replacing good CIFAR-10 synthetic samples with bad ones. Only symPrecision and Clipped Density succeed.

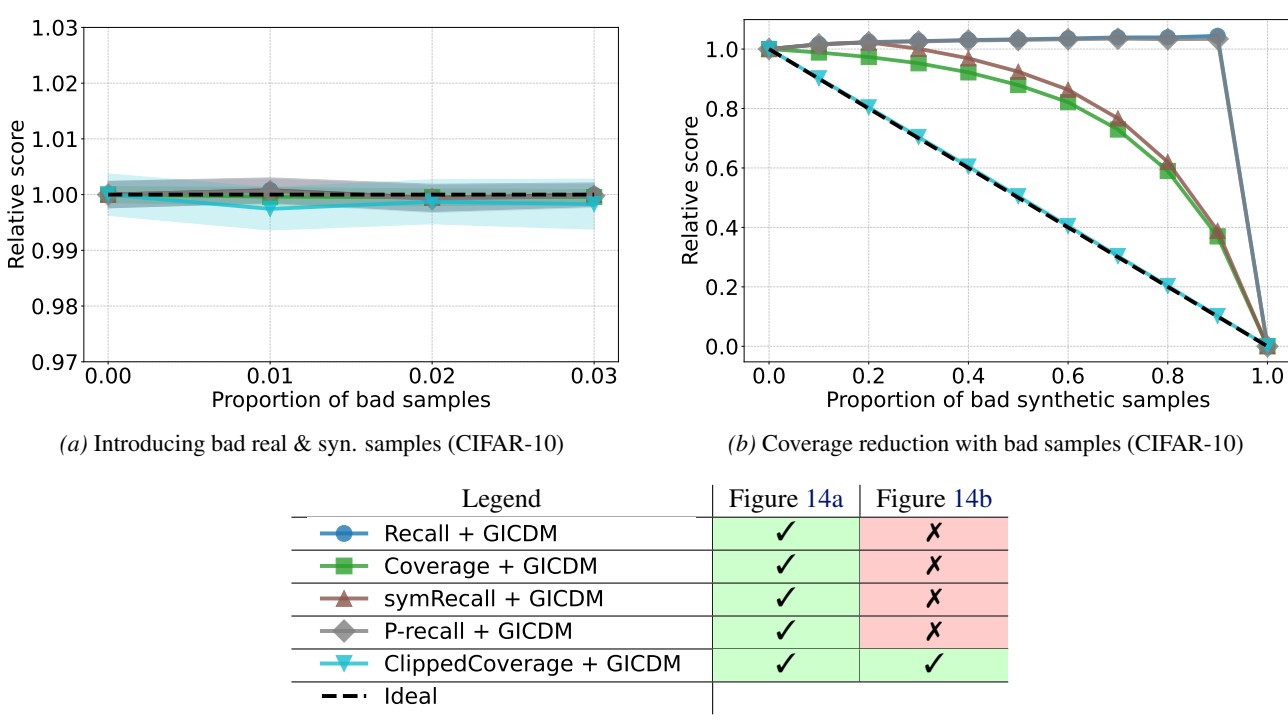

*(a)* Introducing bad real & syn. samples (CIFAR-10)    *(b)* Coverage reduction with bad samples (CIFAR-10)

| Legend | Figure 14a | Figure 14b |
|---|:---:|:---:|
| Recall + GICDM | ✓ | ✗ |
| Coverage + GICDM | ✓ | ✗ |
| symRecall + GICDM | ✓ | ✗ |
| P-recall + GICDM | ✓ | ✗ |
| ClippedCoverage + GICDM | ✓ | ✓ |
| Ideal | | |

*(c)* Legend and summary

*Figure 14.* **Real data tests for coverage metrics**. Coverage metrics with GICDM are evaluated in several scenarios, analogous to those in Figure 13. (a) Introducing real and synthetic out-of-distribution samples at equal rates: all metrics succeed. (b) Progressively replacing good CIFAR-10 synthetic samples with bad ones. Results are consistent with those reported without GICDM: only Clipped Coverage succeeds (Salvy et al., 2026).

previously all fidelity metrics succeeded (Salvy et al., 2026). This failure occurs for the same metrics that failed in the human correlation study in Appendix E.3, specifically those lacking robustness to real outliers (Salvy et al., 2026).

## G. Visualization of hubness effects and GICDM correction

For LSUN Bedroom data embedded with DINOv2, Figure 15 (top row) displays the real image that appears most frequently among the 5-nearest-neighbors of other real images, a *hub*, and the set of real images that include it as a neighbor. This hub is present in 62 neighborhoods, highlighting the severity of hubness. After applying ICDM (Jégou et al., 2010), only 5 real images retain this hub as a neighbor.

The bottom row of Figure 15 presents the same analysis for generated images (from ADM-dropout). The most frequent generated hub is included in the neighborhoods of 64 real images, but after applying GICDM, this number drops to 4.

In both cases, before hubness reduction, many of the images listing the hub as a neighbor are not strongly semantically related to it. After hubness reduction, the remaining images are more semantically similar to the hub image.

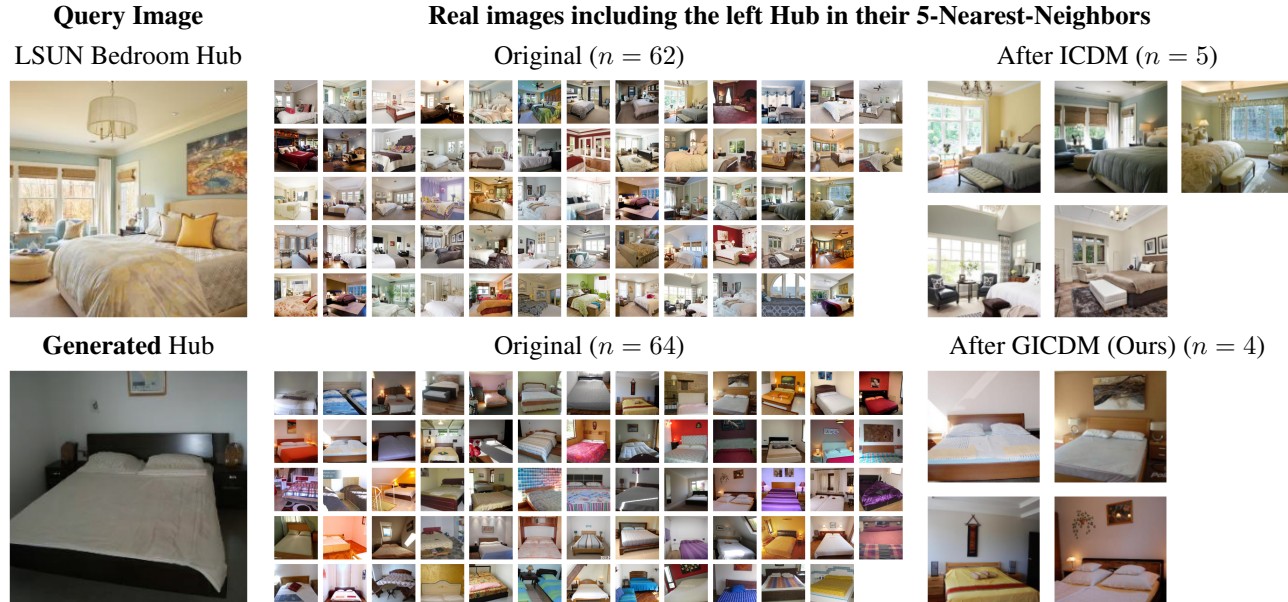

*Figure 15.* **Hubness in DINOv2-embedded LSUN Bedroom and generated sets.** The first column shows the most occurring real (top) and generated (bottom, ADM-dropout) images within the 5-nearest neighborhoods of real images. The middle column displays the real images that include them as neighbors. In both cases, there are more than 60 such real images, identifying the images in the first column as *hubs* and highlighting the asymmetry of neighborhood relationships. After applying hubness reduction (ICDM for real data, GICDM for generated data), the number of real images including these hubs in their neighborhoods drops significantly to 5 and 4, respectively. In both cases, the remaining images listing the hub as a neighbor are more semantically related to it than many of the original ones.

# H. Complexity and Computation time

Let $N$ be the number of real samples and $M$ be the number of generated samples. Computing pairwise distances takes $O(N^2)$ (real-real) and $O(NM)$ (real-generated). Given all pairwise distances, finding the $k$-NN for each sample matches this cost. ICDM repeats this process $T$ times. Once these distances are computed, the remaining operations (computing $\delta$, ratios, and the threshold) are linear. Thus, the overall computational complexity of GICDM is $O(N(TN + M))$.

Note that standard distance-based metrics also scale as $O(N^2)$ in high dimensions because the most efficient method for finding the $k$-NN of all $N$ points is a brute-force approach (Komarov et al., 2013), which constructs the distance matrix and identifies the smallest $k$ elements in each row. Both steps operate in $O(N^2)$ time.

As mentioned in Section 6, all experiments were conducted on a single H100 GPU with 80GB of memory.

- For $M = 50000$ and $N = 50000$, computing scores for all metrics discussed in this work takes approximately 30 minutes (GICDM only needs to be run once). 42 generated sets are evaluated across 4 different embedding spaces for a total of about 84 hours.

- Embedding 50000 samples takes approximately 4 minutes for Inceptionv3 and DINOv2, 5 minutes for VGG16, and 54 minutes for DINOv3. Performing this for 42 generated sets and 4 real sets totals around 51 hours.

- Computing all hubness reduction metrics across all tested parameter values takes on average 1.5 hours per embedding-dataset pair. Repeating this for 17 pairs totals about 25 hours.

- The real data benchmark in Appendix F required around 50 hours of compute, while the synthetic benchmark in Appendix D took roughly 25 hours.

The runtime of the remaining experiments is negligible compared to those stated above. In total, the computation time required to reproduce the results in this paper is approximately 250 GPU hours, while the entire research project (including preliminary experiments) used around 500 GPU hours.

For reference, although we had access to the generated samples from Stein et al. (2023), generating 50000 samples on a single H100 GPU takes approximately 70 hours with ADM (Dhariwal & Nichol, 2021) and about 3 hours with recent latent diffusion models.

# I. Hypersphere test results for each metric

Figure 16 shows the individual results of standard distance-based metrics on the hypersphere test from Figure 1, while Figure 17 displays the results after applying GICDM. Without GICDM, all metrics fail the test, whereas with GICDM, all metrics remain at 0, demonstrating that GICDM effectively resolves hubness-related failures.

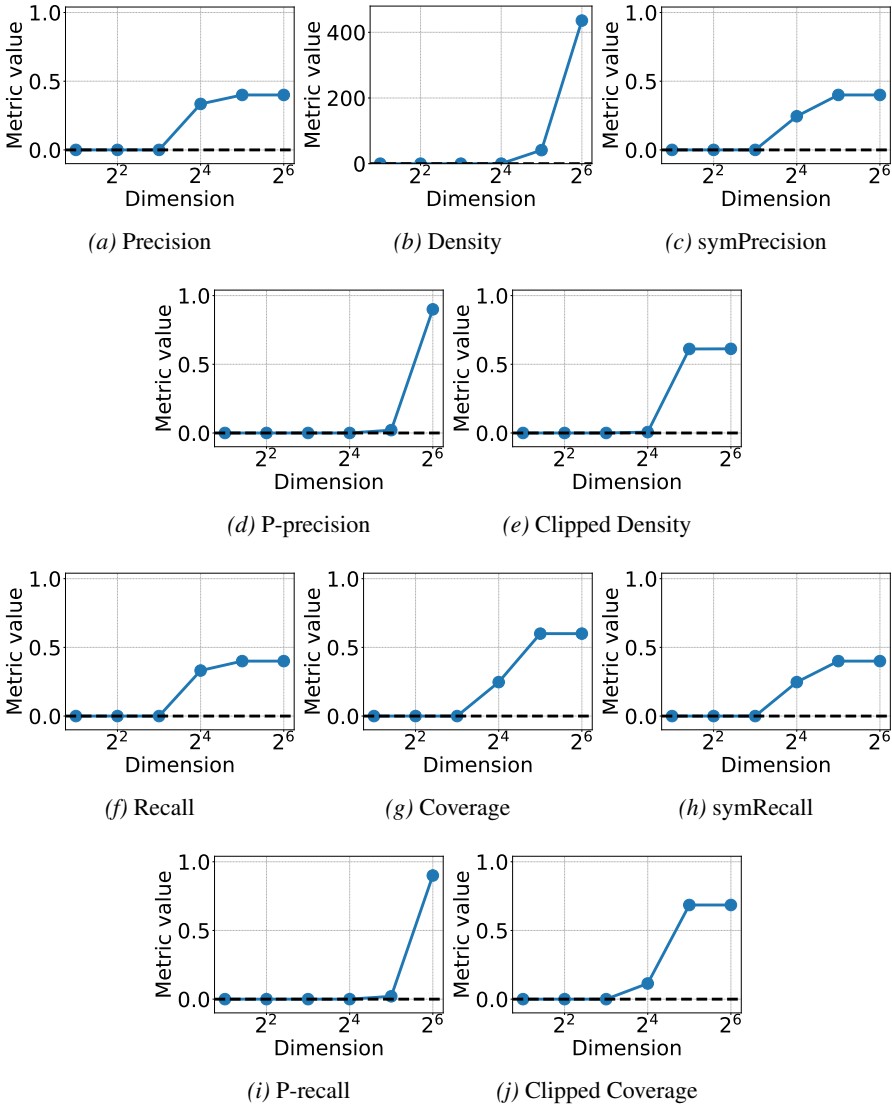

*Figure 16.* **Metrics without GICDM on the Hypersphere Test**: Each subplot shows how a standard metric behaves as the dimension increases in the hypersphere test scenario described in Figure 1. Ideally, all metrics should remain at zero for all dimensions, since the real and generated distributions are disjoint.

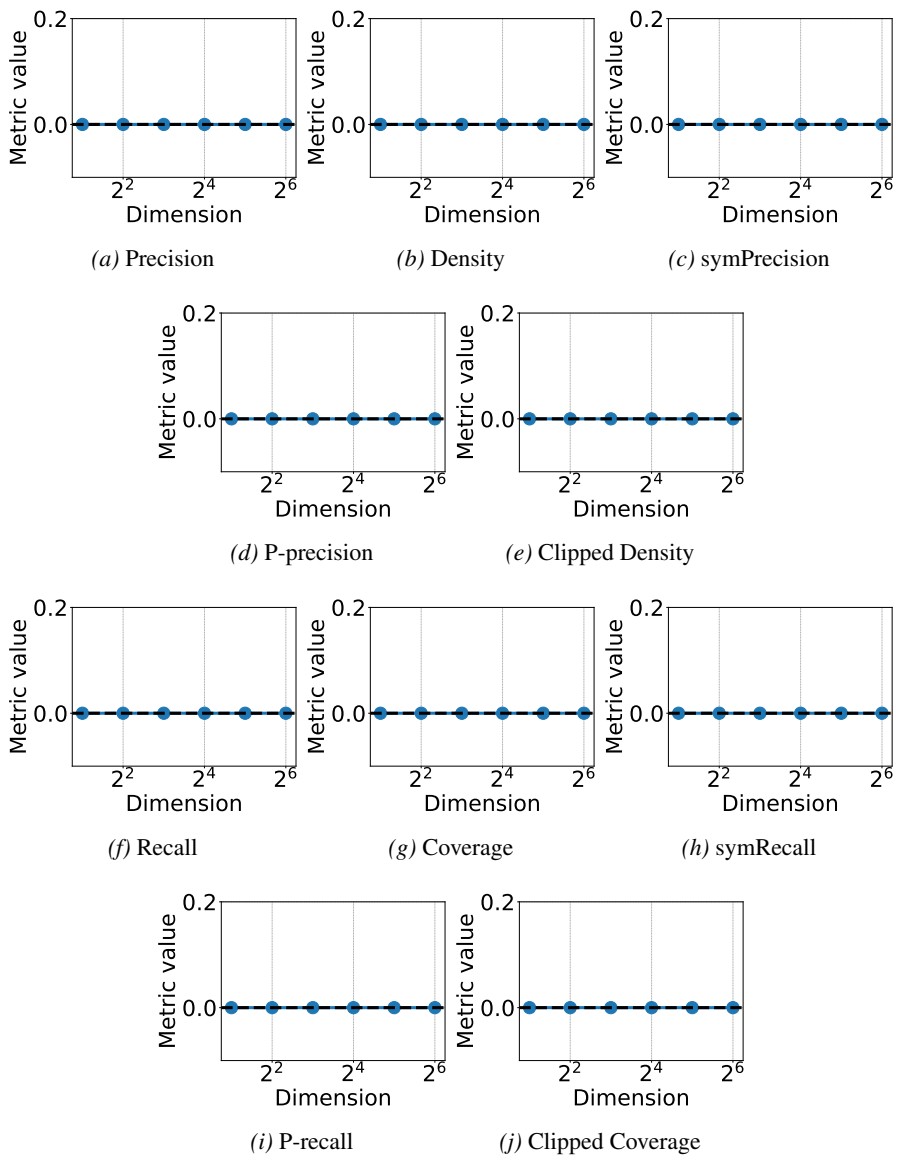

*Figure 17.* **Metrics with GICDM on the Hypersphere Test**: Each subplot shows how a metric, after applying GICDM, behaves as the dimension increases in the hypersphere scenario described in Figure 1. GICDM successfully corrects hubness-related failures, keeping metric values at 0.

## J. In-sample hubness reduction full results

For all hubness reduction methods described in Section 3, as well as projection onto a sphere, Tables 9 and 10 present detailed results for each image dataset and embedding, while Table 11 does so for a music dataset. For methods requiring a neighborhood size $K$, we report the $K$ value in $\{5, 10, 20, 50\}$ that yields the lowest $h_1^5(1\%)$. Results are sorted by dataset and then by $h_1^5(1\%)$.

ICDM consistently achieves the lowest hubness scores across all datasets and embeddings, with optimal $K$ values that are very stable (always 20, except for Inceptionv3 on ImageNet, where it is 10). Its $h_1^5(1\%)$ values are always below 2.0 and $A^5$ remains close to 0.00, indicating that hubness is effectively eliminated. Some hubness reduction methods can occasionally perform worse than no correction, particularly on DINOv2 ImageNet, where DSL, LS, and Sphere projection all increase $h_1^5(1\%)$.

To the best of our knowledge, this is the first demonstration of the capabilities of ICDM in that context.

*Table 9.* **Hubness reduction comparison**: using either DINOv2 (left) or DINOv3 (right). The metrics are $h_1^5(1\%)$, which measures how much more frequently the top 1% of hubs appear in the 5-nearest neighborhoods compared to the average, and $A^5$, the proportion of antihubs. In both cases, lower values are better. On ImageNet with DINOv2 embeddings, some methods perform worse than no correction.

| | DINOv2 EMBEDDINGS | | | DINOv3 EMBEDDINGS | | |
|---|---|---|---|---|---|---|
| DATASET | METHOD | K | $h_1^5(1\%)$ | $A^5$ | METHOD | K | $h_1^5(1\%)$ | $A^5$ |
| CIFAR10 | ORIGINAL | – | 6.5 | 0.14 | ORIGINAL | – | 9.0 | 0.24 |
| | SPHERE | – | 6.2 | 0.12 | SPHERE | – | 6.1 | 0.12 |
| | MP$^{\text{GAUSS}}$ | – | 4.5 | 0.08 | MP$^{\text{GAUSS}}$ | – | 5.0 | 0.10 |
| | LS | 10 | 4.0 | 0.03 | LS | 5 | 4.5 | 0.05 |
| | CSLS | 20 | 2.9 | 0.02 | CSLS | 20 | 3.4 | 0.05 |
| | NICDM | 20 | 2.8 | 0.02 | NICDM | 20 | 3.2 | 0.04 |
| | DSL | 50 | 2.8 | 0.01 | DSL | 50 | 3.1 | 0.01 |
| | ICDM | 20 | 1.7 | 0.00 | ICDM | 20 | 1.8 | 0.00 |
| FFHQ | ORIGINAL | – | 5.8 | 0.12 | ORIGINAL | – | 10.1 | 0.26 |
| | SPHERE | – | 5.5 | 0.11 | DSL | 50 | 6.3 | 0.02 |
| | MP$^{\text{GAUSS}}$ | – | 4.5 | 0.07 | SPHERE | – | 5.6 | 0.10 |
| | LS | 10 | 3.6 | 0.02 | MP$^{\text{GAUSS}}$ | – | 4.9 | 0.10 |
| | DSL | 50 | 2.8 | 0.01 | LS | 20 | 4.2 | 0.07 |
| | CSLS | 20 | 2.6 | 0.02 | CSLS | 20 | 3.6 | 0.06 |
| | NICDM | 20 | 2.6 | 0.02 | NICDM | 20 | 3.3 | 0.05 |
| | ICDM | 20 | 1.7 | 0.00 | ICDM | 20 | 1.8 | 0.00 |
| LSUN BEDROOM | SPHERE | – | 6.0 | 0.14 | ORIGINAL | – | 9.0 | 0.23 |
| | ORIGINAL | – | 5.8 | 0.13 | SPHERE | – | 7.0 | 0.15 |
| | MP$^{\text{GAUSS}}$ | – | 4.7 | 0.09 | MP$^{\text{GAUSS}}$ | – | 5.4 | 0.12 |
| | LS | 20 | 3.7 | 0.03 | LS | 20 | 4.2 | 0.07 |
| | DSL | 50 | 3.0 | 0.01 | DSL | 50 | 3.5 | 0.01 |
| | CSLS | 20 | 2.7 | 0.02 | CSLS | 20 | 3.4 | 0.06 |
| | NICDM | 20 | 2.7 | 0.02 | NICDM | 10 | 3.2 | 0.04 |
| | ICDM | 20 | 1.7 | 0.00 | ICDM | 20 | 1.8 | 0.00 |
| IMAGENET | DSL | 5 | 7.3 | 0.01 | ORIGINAL | – | 8.0 | 0.20 |
| | LS | 5 | 5.5 | 0.02 | LS | 50 | 5.0 | 0.14 |
| | SPHERE | – | 4.4 | 0.10 | SPHERE | – | 4.7 | 0.12 |
| | ORIGINAL | – | 4.4 | 0.10 | MP$^{\text{GAUSS}}$ | – | 4.0 | 0.10 |
| | MP$^{\text{GAUSS}}$ | – | 2.8 | 0.03 | CSLS | 20 | 3.3 | 0.06 |
| | CSLS | 10 | 2.6 | 0.01 | DSL | 20 | 3.1 | 0.01 |
| | NICDM | 10 | 2.5 | 0.01 | NICDM | 20 | 3.1 | 0.05 |
| | ICDM | 20 | 1.8 | 0.00 | ICDM | 20 | 1.9 | 0.00 |

*Table 10.* **Hubness reduction comparison**: using either Inceptionv3 (left) or VGG16 (right) embeddings. The metrics and datasets are the same as in Table 9. The initial hubness levels are generally higher than with DINO embeddings, but ICDM still consistently achieves the lowest hubness scores across all datasets and embeddings.

| DATASET | INCEPTIONV3 EMBEDDINGS | | | | VGG16 EMBEDDINGS | | | |
|---|---|---|---|---|---|---|---|---|
| | METHOD | K | $h_1^5(1\%)$ | $A^5$ | METHOD | K | $h_1^5(1\%)$ | $A^5$ |
| CIFAR10 | ORIGINAL | – | 9.0 | 0.22 | ORIGINAL | – | 10.4 | 0.21 |
| | SPHERE | – | 6.2 | 0.12 | SPHERE | – | 5.6 | 0.10 |
| | MP$^{\text{GAUSS}}$ | – | 4.5 | 0.07 | MP$^{\text{GAUSS}}$ | – | 4.3 | 0.06 |
| | LS | 20 | 3.8 | 0.04 | LS | 50 | 3.7 | 0.04 |
| | CSLS | 50 | 3.3 | 0.04 | CSLS | 50 | 3.3 | 0.04 |
| | NICDM | 20 | 3.2 | 0.03 | NICDM | 20 | 3.2 | 0.03 |
| | DSL | 50 | 2.9 | 0.01 | DSL | 50 | 2.4 | 0.01 |
| | ICDM | 20 | 1.7 | 0.00 | ICDM | 20 | 1.7 | 0.00 |
| FFHQ | ORIGINAL | – | 8.3 | 0.20 | ORIGINAL | – | 9.8 | 0.23 |
| | SPHERE | – | 8.1 | 0.17 | SPHERE | – | 6.2 | 0.12 |
| | MP$^{\text{GAUSS}}$ | – | 4.9 | 0.08 | MP$^{\text{GAUSS}}$ | – | 4.8 | 0.08 |
| | LS | 20 | 3.9 | 0.04 | LS | 20 | 3.7 | 0.04 |
| | CSLS | 50 | 3.2 | 0.04 | CSLS | 20 | 3.3 | 0.04 |
| | NICDM | 20 | 3.1 | 0.03 | NICDM | 20 | 3.2 | 0.03 |
| | DSL | 50 | 2.6 | 0.01 | DSL | 50 | 2.5 | 0.01 |
| | ICDM | 20 | 1.7 | 0.00 | ICDM | 20 | 1.7 | 0.00 |
| LSUN BEDROOM | ORIGINAL | – | 12.9 | 0.28 | ORIGINAL | – | 16.6 | 0.31 |
| | SPHERE | – | 9.0 | 0.19 | SPHERE | – | 7.0 | 0.11 |
| | MP$^{\text{GAUSS}}$ | – | 5.5 | 0.10 | MP$^{\text{GAUSS}}$ | – | 4.6 | 0.07 |
| | LS | 20 | 4.1 | 0.05 | LS | 50 | 4.3 | 0.06 |
| | CSLS | 50 | 3.8 | 0.06 | CSLS | 50 | 4.0 | 0.06 |
| | NICDM | 20 | 3.6 | 0.04 | NICDM | 20 | 3.8 | 0.04 |
| | DSL | 50 | 3.1 | 0.01 | DSL | 50 | 2.4 | 0.01 |
| | ICDM | 20 | 1.7 | 0.00 | ICDM | 20 | 1.7 | 0.00 |
| IMAGENET | ORIGINAL | – | 5.7 | 0.20 | ORIGINAL | – | 12.6 | 0.20 |
| | SPHERE | – | 4.7 | 0.11 | SPHERE | – | 4.5 | 0.11 |
| | LS | 50 | 4.7 | 0.09 | MP$^{\text{GAUSS}}$ | – | 3.9 | 0.07 |
| | MP$^{\text{GAUSS}}$ | – | 4.0 | 0.08 | LS | 20 | 3.9 | 0.05 |
| | DSL | 20 | 3.4 | 0.01 | CSLS | 20 | 3.2 | 0.04 |
| | CSLS | 20 | 3.0 | 0.04 | NICDM | 20 | 3.1 | 0.03 |
| | NICDM | 20 | 2.9 | 0.03 | DSL | 20 | 2.8 | 0.00 |
| | ICDM | 10 | 1.7 | 0.00 | ICDM | 20 | 1.7 | 0.00 |

*Table 11.* **Hubness reduction comparison**: for CLAP embeddings of 7-second music samples from the Free Music Archive (FMA) dataset. The metrics are the same as in Tables 9 and 10. Even in this different modality, hubness is present, and ICDM achieves the best performance.

| DATASET | CLAP EMBEDDINGS | | | |
|---|---|---|---|---|
| | METHOD | K | $h_1^5(1\%)$ | $A^5$ |
| FMA | ORIGINAL | – | 4.9 | 0.09 |
| | SPHERE | – | 4.6 | 0.08 |
| | MP$^{\text{GAUSS}}$ | – | 3.9 | 0.06 |
| | LS | 20 | 3.3 | 0.02 |
| | CSLS | 20 | 2.6 | 0.01 |
| | NICDM | 20 | 2.5 | 0.01 |
| | DSL | 50 | 2.4 | 0.01 |
| | ICDM | 20 | 1.8 | 0.00 |

