# OpenReview forum: "GICDM: Mitigating Hubness for Reliable Distance-Based Generative Model Evaluation"
_ICML.cc/2026/Conference — ICML 2026 regular_

### Official Review · Reviewer_6sH2 · 2026-02-14

**Soundness:** 3
**Presentation:** 3
**Significance:** 4
**Originality:** 3
**Overall Recommendation:** 5
**Confidence:** 3

**Summary:**

The paper describes hubness as an issue in high-dimensional embedding spaces that will alter the performance of many popular evaluation metrics.

The paper proposes GICDM, a new algorithm for measuring distance in embedding space that reduces hubness. The resulting neighborhoods from this method can be used as input to existing evaluation metrics. The paper provides theoretical results describing the behavior of GICDM and justifications for choice of certain hyperparameters (e.g. size of neighborhood based on cross-over dimension), and empirical results on synthetic and real datasets from popular image generative models.

**Compliance With Llm Reviewing Policy:**

Affirmed.

**Final Justification:**

The authors have addressed the majority of my concerns, and so I have decided to raise my final score from a 4 to a 5 given that this paper is technically sound and provides some technical advancements on a problem of interest to the community.

All of my questions about the paper have been thoughtfully addressed by the authors.

Soundness: I originally provided a score of 2 due to some unclear notions of how GICDM works. After clarifications during the rebuttal, I have raised this score to a 3.  Given that this method does not always work (In Table 6, some metrics suffer after applying GICDM), I cannot state that this method deserves a soundness of 4.

Presentation: The paper is overall well-written. My review and others have presented numerous areas of confusion, but the authors have acknowledged these issues.

Significance: Evaluating generative models is extremely important, and the design of useful metrics is an active area of research. This paper suggests that GICDM can improve many of these workflows. I originally gave a score of 4, and am willing to maintain this rating. GICDM is not a perfect catch-all, but GICDM has the potential to have significant impacts on lots of future research in this area. I was concerned with the general utility of GICDM due to its run-time, but the authors have provided sufficient evidence that even if GICDM is slow, it is not prohibitively slow that it cannot be deployed.

Originality: Hubness has been studied previously, but extending it to this exact problem space is novel.

Overall, I thank the authors for their thorough responses and support this paper's potential acceptance.

**Key Questions For Authors:**

1) My main concern: The underlying distance measure in this paper is never made clear. Normally, people just measure cosine similarity on normalized embeddings. In this setting, the idea of hubness being samples close to the origin doesn't seem to apply. In Line 154, the authors claim that: 'in a non-gaussian setting, simply projecting points onto a sphere does not eliminate hubness'. Can you provide more evidence for this claim (or point to a part of the paper where this evidence is)?

2) In Algorithm 1 and section 4.3, you mention that generated points that are too far from the training distribution (based on quantiles) are discarded. Are these samples just ignored then and not included in the metric computation? Is this a reasonable approach to take? Intuitively, removing outliers from a generative model will bias any evaluations you make of it.

3) Can you outline the overall time complexity of this approach? Can you compute this efficiently on larger datasets? E.g. for the Stein et al. experiments, only 50,000 real images are used. This is a small fraction of larger datasets like ImageNet1K. I am not asking you to re-run the experiments on that larger scale, but it does seem like a limitation of this approach. Is computing other metrics like ClippedDensity equivalent complexity? In that case, the hubness approach would not be the sole bottleneck.

4) Many of the evaluated generative models are not recent (e.g. 2023 and before). While adding new models to this benchmark may not be feasible, I am curious to know what would happen if you evaluated on more recent model.

5) How should I interpret the significance of figures 8 and 9? Is increasing the spread of scores a useful result?

**Limitations:**

yes

**Strengths And Weaknesses:**

This paper aims to tackle a major limitation in how we evaluate generative models with respect to their training data.

The paper does a great job of providing background into related methods, how they work, what they were used for, and how they plan on building on them throughout the rest of the paper.

The description of the GICDM algorithm is also laid out well. The authors also provide a wide array of experiment to prove the effectiveness of their algorithm.

My main concerns about the paper are outlined in the questions section. While the authors have provided convincing evidence that hubness is a problem for generative models, it is not entirely clear to me that GICDM provides a sufficient solution. I am primarily concerned with 1) The type of distance metric used, if hubness is a problem if they had used a different metric, and if GICDM works under those metrics 2) If the removal of outliers is biasing results in some ways.

With satisfactory clarifications about the core contributions of their method, I would raise the rating.

---

> ### Author Rebuttal · Authors · 2026-03-30
>
> We thank the reviewer for their positive feedback on the paper’s presentation and significance. We address your comments follow:
> ## Projection on a Sphere
> >Q1. Provide evidence that projecting onto a sphere does not eliminate hubness
>
> Table 5 compares hubness reduction methods, including projection onto a sphere ("Sphere") which performs better than nothing, but underperforms all other methods. We will reference Table 5 on L154R.
>
> Hubness is linked to density gradients (pointing towards the center for Gaussians). Fig. 4a and 4b show 2D examples with hubness using the Euclidean distance. Adding a small curvature to project them onto a small portion of a 3D sphere (picture one panel of a football), will keep hubness. The local cosine similarity mimics Euclidean distance, preserving density gradients/boundaries and thus hubness. Eliminating hubness on a sphere requires uniform density.
>
> Feldbauer et al. (2019) "do not recommend" simply changing the distance as it is not competitive with the best hubness reduction methods. Note that GICDM can work with other distances or dissimilarities like the cosine similarity.
> ## No sample is removed
> >Q2. Is it reasonable to discard samples?
>
> This is a misunderstanding caused by our lack of explanation of what "filter out" means L323.
> We will clarify 328:
> >>**These filtered points fall outside the real data manifold. As they are in no real ball, their fidelity score is zero and they do not contribute to coverage.**
>
> These samples **are** evaluated: scoring 0 fidelity and adding 0 coverage.
>
> Ignoring some samples would indeed bias evaluations. This happens without hubness reduction due to antihubs (samples with occurrence $O_k(x)=0$). Generated samples in antihub regions aren't effectively evaluated. A 0 score can mean a bad generated sample, or simply a generated antihub.
> ## Complexity
> >3. What is the complexity, is it a new bottleneck?
>
> The complexity in the number of samples of GICDM and metrics is the same.
>
> Computing pairwise distances takes $O(N^2)$ (real-real) and $O(NM)$ (real-generated). Given all pairwise distances, finding the $k$-NN of each sample matches this cost. ICDM repeats this $T$ times. Given those values, everything else (computing $\delta$, ratios, threshold) is linear, so overall the complexity is $O(N(TN + M))$.
>
> **Metrics also run in $O(N^2)$ in high dimension**. Indeed, in high dimensional spaces the most efficient method for finding the $k$-NN of all $N$ points is the brute force method ([I. Komarov et al., 2014](https://arxiv.org/pdf/1309.5478)) which builds the distance matrix and finds the smallest $k$ elements of each row. Both of these steps are in $O(N^2)$. This explains why usually a maximum of $50000$ samples are used for evaluation.
> ## Evaluating a recent model
> >4. What would happen if you evaluated on more recent model?
>
> We evaluated data generated by a modern latent diffusion model on ImageNet: LightningDiT in a VA-VAE (J. Yao et al., 2025), with and without classifier-free guidance. Results are as follows:
>
> In DINOv2:
> |Model|Clipped Density + GICDM|Clipped Coverage + GICDM|
> |-|-|-|
> |LDM|0.81|0.69|
> |DiT-XL-2|0.86|0.79|
> |DiT-XL-2-guided|1.15|0.55|
> |VAVAE-LightningDiT (no-cfg)|0.73|0.79|
> |VAVAE-LightningDiT (cfg)|0.89|0.88|
>
> In DINOv3:
> |Model|Clipped Density + GICDM|Clipped Coverage + GICDM|
> |-|-|-|
> |LDM|0.45|0.42|
> |DiT-XL-2|0.50|0.47|
> |DiT-XL-2-guided|0.94|0.53|
> |VAVAE-LightningDiT (no-cfg)|0.47|0.53|
> |VAVAE-LightningDiT (cfg)|0.60|0.62|
>
> VAVAE-LightningDiT with classifier-free guidance achieves the best coverage score with both embeddings. Note that they use cfg interval ([T. Kynkäänniemi et al., 2024](https://arxiv.org/pdf/2404.07724)) which might explain the improvement to both fidelity and coverage.
>
> Overall, VAVAE-LightningDiT seems to outperform the older latent diffusion model LDM, but the pixel-space diffusion model DiT-XL-2 still generates higher-fidelity images (with worse coverage).
> ## Figures 8-9
> >5. How should I interpret the significance of figures 8 and 9? Is increasing the spread of scores a useful result?
>
> In general having a larger spread indicates a better discriminative power. In this case, we have no ground truth for the absolute values of the metrics. We can only conclude that this behavior is better thanks to the improved correlation with human error rates in Figures 10 and 11.
>
> Figures 8 and 9 serve more as an updated benchmark of the evaluated generative models. In particular, with DINOv2, it recovers that basic classifier-free guidance improves fidelity at the cost of coverage.
> ## References not in the paper
> I. Komarov et al., "Fast k-NNG construction with GPU-based quick multi-select." *PloS one* 9.5, 2014
>
> T. Kynkäänniemi et al., "Applying Guidance in a Limited Interval Improves Sample and Distribution Quality in Diffusion Models", NeurIPS 2024
>
> J. Yao et al., "Reconstruction vs. generation: Taming optimization dilemma in latent diffusion models", CVPR 2025

---

> > ### Author Rebuttal · Reviewer_6sH2 · 2026-04-02
> >
> > The authors have addressed many of my concerns However, upon reading other reviews and further consideration, I have some new questions that I would like clarity on before I adjust my rating.
> >
> > ### Scaling of GICDM
> >
> > In all of the Stein et. al experiments, you use 50,000 sampled images due to issues with how GICDM scales. While the listed evaluation metrics are also O(N^2), there are many other popular evaluation metrics for generative models that are faster. For example, Frechet Distance (FD) is O(d^3), where d is the dimension of the data, and was shown to correlate well with human evaluation in Stein et al. (Figure 4 in their paper).
> >
> > On experiments on ImageNet, sampling only 50,000 images (or 50 images per class) will assuredly introduce noise into your metric computation. This level of noise can easily drown out any improvements that are made from GICDM.
> >
> > This seems like a core limitation of this approach and does limit its broader applicability.
> >
> > ### Loss of information after removing hubness.
> >
> > Having metrics align better with human judgement is not necessarily equivalent to 'no information was lost'. The paper generally focuses on precision and density. There are many other aspects of generative model evaluation beyond these two approaches. I believe this work has provided sufficient evidence that it can improve the utility of ClippedDensity and ClippedCoverage, but these metrics are not exhaustive and it is possible that there are other metrics that will be less reliable after GICDM is applied.
> >
> > I do maintain an overall positive opinion of this work, but I would like some clarity regarding the significance of my work before adjusting my rating.

---

> > > ### Author Response · Authors · 2026-04-03
> > >
> > > We thank you the thoroughness of your comments.
> > > ## Scaling of GICDM
> > > > using 50,000 samples will introduce noise
> > >
> > > The computation time of GICDM is not a limitation in its applicability. We only used 50000 samples because:
> > > - For some generated sets, we only have access to 50000 samples;
> > > - **Generating** (rather than evaluating) this many samples can be expensive (see below);
> > > - Using 50000 samples is common practice and sometimes only 10000 samples are used (e.g., J. Yao et al., 2025 report some FID-10k and some FID-50k). Evaluation using more than 50000 generated samples appears unusual.
> > >
> > > **Stein et al. also use 50000 generated samples** for their FD evaluation (see their Apdx B.1).
> > >
> > > While FD indeed runs faster, evaluation is not a bottleneck in practice. For reference:
> > > - **Training** LightningDiT for 800 epochs takes around 125 hours on 8xH800 GPUs (see https://github.com/hustvl/LightningDiT), not including the variational autoencoder training time.
> > > - **Generating** 50000 samples on 1 H100 GPU takes around:
> > >   - 68 hours with ADM (Dhariwal & Nichol, 2021).
> > >   - 3 hours with the most recent model we evaluated (LightningDiT in a VA-VAE, J. Yao et al., 2025).
> > > - **Embedding** 50000 samples on 1 H100 GPU takes around:
> > >   - 5 minutes in DINOv2.
> > >   - 55 minutes in DINOv3.
> > > - **Evaluating** 50000 samples on 1 H100 GPU takes on average around:
> > >   - 30 seconds with FD.
> > >   - 2 minutes for both scores from Clipped Density and Clipped Coverage
> > >   - 21 minutes for both scores from Clipped Density + GICDM and Clipped Coverage + GICDM (which is around 10 times the time taken without GICDM running for t=10 iterations).
> > >   - 30 minutes to get scores from all metrics cited in our work on one set (GICDM only needs to run once).
> > >
> > > Thus, the computing time of GICDM is really quite neglibible compared with training and generating times.
> > >
> > > Furthermore, using 50000 real samples does not result in a large amount of noise. We evaluated DiT-XL-2 using 10 disjoint 50,000-sample subsets of ImageNet. The results are highly stable:
> > > - DINOv2: Clipped Density + GICDM ranged from 0.858 to 0.869; Clipped Coverage + GICDM from 0.785 to 0.792.
> > > - DINOv3: Clipped Density + GICDM ranged from 0.497 to 0.510; Clipped Coverage + GICDM from 0.471 to 0.477.
> > >
> > > The maximum variation of roughly 0.01 is too small to drown out GICDM’s improvements. This aligns with Salvy et al. (2026, Appendix M), who found that the standard deviation of these metrics scales as $\frac{1}{\sqrt{N}}$ "reaching values below 0.01 for $N=50000$".
> > > ## Loss of information after removing hubness.
> > > >Information loss? Other metrics?
> > >
> > > We agree that aligning better with human judgement does not mean no information is lost. It simply ensures the evaluation "makes sense". Proving metrics measure the correct properties without ignoring others is hard, hence why metrics are subjected to extensive empirical testing.
> > >
> > > We focus on **fidelity** and **coverage**. Common metrics either measure those two aspects (e.g. Fidelity&Recall, Density&Coverage, etc.) or measure both aspects at the same time (e.g. FID, FD-DINOv2, CMMD). There are indeed other aspects of data generation that should be evaluated, such as memorization (are generated samples just very close reproductions of the training data?) and we acknowledge we do not tackle this particular issue.
> > >
> > > We also agree that testing many metrics is important to prove GICDM is not specific to Clipped Density/Coverage.  Because of space constraints the main text highlighted those two as they performed best, but we actually evaluated >10 different metrics (Apdx C, D, and E).
> > >
> > > In particular, on the full Räisä 2025 benchmark (Apdx C, Tables 6-7), GICDM’s improvements are not limited to those two metrics. We found (L947-948) that "GICDM consistently improves Purpose results (shape of the curve) on both the ”Gaussian Std. Deviation Difference” and ”Hypersphere Surface” tests".
> > >
> > > We will expand our Experiments section to clarify this L416-422 left (changes in bold):
> > > >>Synthetic Benchmark. We evaluated standard metrics and their GICDM-corrected versions on the **extensive** synthetic benchmark **proposed by**  Räisä et al. (2025). **This benchmark evaluates the evolution of metric results across 14 *Purpose* scenarios, tests whether metrics yield the expected values (e.g., 1 for realistic generated data) in 13 *Bounds* cases, and checks 3 *Other* miscellaneous tests. We ran this benchmark on more than 10 metrics (with full details in Appendix C), and found that GICDM consistently improves two Purpose tests.** Table 1 **summarizes the results for the best-performing fidelity and coverage metrics: applying** GICDM **substantially** improves performance, **passing 23 criteria instead of 18 for Clipped Density with GICDM, and 22 instead of 18 for Clipped Coverage with GICDM. This underlines** that hubness **is** indeed the root cause of several failures **and that GICDM successfully corrects this issue without altering performance on criteria that were already satisfied.**

---

### Official Review · Reviewer_9L4p · 2026-02-20

**Soundness:** 3
**Presentation:** 2
**Significance:** 2
**Originality:** 2
**Overall Recommendation:** 3
**Confidence:** 3

**Summary:**

The paper studies the effects of the hubness phenomenon on evaluating generated images in generative deep learning models. Hubs are specific points frequently appearing in kNN lists of several other points in a high-dimensional data set. These hubs in space notion is similar to hub nodes on social network graphs, but defined on kNN graphs.
As the current methods used on the generative model evaluation are based on kNN neighborhoods between generative images and real images, there are potential effects by these hubs, i.e.  kNN between real and generative images tend to be skew towards a few hub images. The paper proposes the method called GICDM that uses a variant of the weighted dissimilarity $d(x_i, x_j) / \mu_i \mu_j$ where the weight $\mu_i, \mu_j$ is derived from the average kNN distance of real images $x_i$ and $x_j$ over the real dataset $X$.
The experiment shows that the combination of GICDM with the current Clipped Density and Coverage (Salvy et al. ICLR 26) improves the correlation with human scores on popular data sets.

**Compliance With Llm Reviewing Policy:**

Affirmed.

**Final Justification:**

__Update after rebuttal, before in bracket (See Rebuttal Acknowledgement for details)__

I have raised the overal recommendation from reject to weak reject but reduced my confidence to 3. While I understand well the contribution in the algorithmic way, I could not see the novelty of the work with respect to generative model evaluation problem and how important it is. I would prefer the discussion with other reviewers for the final decision.

Soundess: 3 (1), Presentation 2 (2), Significance 2 (1), Originality 2 (1)

Overall recommendation: Weak reject (reject)

Confidence: 3 (5)

**Key Questions For Authors:**

Please address the raised weaknesses above and questions below.

Q1) Since the hubs are points that frequently appears in the kNN lists, should we remove them or adjust its contribution in Clipped Density or Coverage scores? Since GICDM runs in $O(n^2)$, which is similar to the brute force solution to count the frequency of a point being kNN lists, I wonder why we would not consider this simpler approach.

Q2) How large skewness of the frequency of the hub that cause unwanted effects? In the appendix A, you state "the top 1% most frequent points appear more than 4 times
as often as the average point among the 5-nearest neighbors of other points, and 10% of points are antihubs, never appearing
in any 5-nearest neighbor list." Would it be normal values on datasets with several dense clusters?

There are several confused notions that need to be clarified:
- crossover dimension $d*$,
- multi-scale filtering with $K_1, K_2$
- L234, L236: large or small $\delta_i$ depends on the relative density, not the absolute high/low density regions
- Fig 1: Hard to understand as there is no description of the generating mechanism.

**Limitations:**

Yes

**Strengths And Weaknesses:**

Although the proposed GICDM combining ICDM with the current Clipped Density and Coverage (Salvy et al. ICLR 26) to improve the correlation with human scores on popular data sets on generative model evaluations, several weaknesses downweight the contributions of the paper.

__W1) Lack of novelty in the proposed method GICDM__

The GICDM was proposed in Jegou et al 2010, as a weighted dissimilarity measure (note that this is not a metric distance). The submission proposes to use this dissimilarity as a distance measure used in computing Clipped Density and Coverage scores (Salvy et al.). As GICDM is not a metric distance, which is different from the metric distances used in Salvy et al. and other works, there should be in-depth discussion regarding the implications of the used non-metric distances on generative model evaluation.

I note that the cost of computing GICDM is $O(tn^2)$, where $t$ is the number of iterations, and $n$ is the size of real dataset, which is much worse than standard distance measures. This limits GICDM on evaluating generative images on large-scale settings.

__W2) Unclear/superficial statements on key desiderate for hubness mitigation__

- L216: The local density of real data manifold should be uniform, ... nearest neighbor relationships are symmetric.

I wonder why we would need these properties when reducing hubness in real data. In principle, we cannot change the manifold distribution of real data, and the nearest neighbor relationships are asymmetric. Forcing nearest neighbor relationships to symmetric would completely change the nearest neighbor notion.

- L220. Preserve the relative positioning of generated points (i.e. maintaining the spatial relationships between generated images and real images)

The spatial relationships are represented by a similarity/dissimilarity measure. Different measures used would lead to different spatial relationships. How would we ensure this property?

- L227. Adding or removing generated images does not affect the evaluation of others.

Since the purpose of generative model evaluation is to evaluate the quality of generated images, and I recall the Clipped Density and Coverage consider the relationships between one generative image and the real dataset, I wonder why we would remove generated images in this case.


I found that the figures used to elaborate the hubness phenomenon is from synthetic Gaussian distribution (Fig 1, 2, 3, 4, 5, 6). However, the generated images are generated from the intrinsic distribution of real data. Therefore, these figures are misleading on reflecting the contribution of the work on improving the generative model evaluations.

__W3) Potential issues on the use of GICDM and proposed methodology__

I found some incorrect arguments.

L264: Since the generated sample $x^g_j$ is drawn from the distribution of real data, we could not have $\mu_j^{g, eq} \approx {\bar{\mu}}^{r, eq}$ for any fixed instance $x^g_j$. We can only have it in expectation. The use of this formula for a specific instance to derive the fomula of $\mu_j^{g, eq}$ on L264 is not correct.


While we would evaluate quality of ALL generated images, the paper proposed to filter out some generated images that are not aligned with the real data manifold based on an assumption (L311, Section 4.3) using the proposed dissimilarity measure. This step would clearly affect the human score-based correlation metric for generative model evaluations in the experiment since you are filtering out generated images in evaluation process. Therefore, the improvement when combining with Clipped Density and Covering are not reliable. Unfortunately, I do not understand multi-scale filtering as I cannot see the multi-scale notion. However, the more generated images you filter out, the more unreliable your generative model evaluation process is.


__Conclusion__

Given the lack of novelty in GICDM, superficial statements on the core properties of the designed method, and potential flaws on the methodology, I would lean on a rejection.

---

> ### Author Rebuttal · Authors · 2026-03-30
>
> Thank you for the detailed review. We address all your comments bellow.
> ## Novelty
> >W1) "GICDM was proposed in Jegou et al 2010"
>
> ICDM (Jegou et al. 2010) works on a *single* dataset. **G**ICDM extends this to a real set and its generated counterpart, allowing generated points to be evaluated against real neighborhoods.
>
> Applying ICDM to the real set alone does not give access to real-generated distances (ICDM yields real-real dissimilarities, not new coordinates). Applying it to their union makes real neighborhood structures dependent on generated points.
>
> GICDM solves this: we derive $\delta^g$ for generated points at ICDM convergence assuming they are realistic (Sec 4.2), then relax this assumption by preventing $\delta^g$ parameters from diverging too much from neighboring $\delta^r$ (Sec. 4.3, 4.4).
> ## Hubness should be corrected / Removing hubs
> >W2) Why symmetrize nearest neighbor relationships?
> >Q1) Why not remove hubs?
>
> Please see our answers to reviewer BTjk, marked with "## Hubness should be corrected" and with "## Removing hubs".
> ## Incorrect arguments
> >W3) incorrect arguments
> ### Derivation of $\mu_j^{g,eq}$
> >L264: cannot have $\mu_j^{g,eq}\approx\bar{\mu}^{r,eq}$ for a fixed instance, only in expectation.
>
> We respectfully point out that $\mu^{eq}$ is not a statistical property of raw data, so **expectation is not the relevant concept here**. For any sample $x^r_i$, $|\mu_i-\bar{\mu}|<\epsilon$ for an arbitrary $\epsilon>0$ used to define convergence. This is not just in expectation, $\mu^{r,eq}$ is **deterministically** obtained at ICDM convergence. We will clarify this L220, 258 and 264.
>
> When assuming that a generated sample $x^g_j$ is drawn from the real distribution (L261-262), its **$\mu^{g,eq}$ is undefined**. To ***set*** it (L262-263), we state "its average neighbor distance should match that of the real samples: $\mu^{g,eq}_j\approx\overline{\mu^{r,eq}}$. **We *enforce* this property.**
>
> The following sections relax this realism assumption.
> ### No sample is removed
> >W3: filtering images makes evaluation unreliable
>
> Please look at our answer to reviewer 6sH2, marked with ## No sample is removed.
> ## Complexity
> >W1
>
> Please look at our answer to reviewer 6sH2, marked with ## Complexity.
> ## Metric distance
> >W1: GICDM is not a metric distance, can it be used for generative model evaluation?
>
> Yes, thank you for raising this point. Indeed, GICDM is not a metric distance but a dissimilarity: it does not follow the triangular inequality.
>
> Distance-based generative model evaluation relies on ball neigborhoods around reference data points, which are sub-level sets of the dissimilarity: point $x^g_j$ is considered in the neighborhood of $x^r_i$ iff $\text{GICDM}(x^r_i,x^g_j)\le\text{NND}_k^r(x^r_i)$ where $\text{NND}\_k^r(x^r_i)$ is the $k$-th smallest dissimilarity scalar $\\{\text{ICDM}(x_i^r,x_l^r)\\}\_{l\neq i}$ in the real set. This only requires ranking (for NND) and thresholding, never the triangular inequality.
>
> In fact, dissimilarities like cosine similarity are standard in these spaces (Oquab et al., 2023; Siméoni et al., 2025). We will include this discussion L367 left.
> ## Figures based on Gaussians
> >W2) Fig 1-6 are misleading
>
> These Gaussian examples illustrate fundamental high-dimensional phenomena:
> - Fig 1 & 6: hypersphere benchmarks where metrics fail due to hubness, and GICDM ablations.
> - Fig 2-4: explain hubness and density gradients.
> - Fig 5: crossover regime dynamics.
>
> Our real-world results (Tables 1-5, 8-11; Figs 8-14) on images and audio validate these insights.
> ## When does hubness become issue
> >Q2) How large must skewness be to cause issues? Are Appendix A values standard?
>
> Even a small proportion of antihubs create blind spots.
>
> We use Skewness ($S^{k=10}$, lower is better) to compare to Feldbauer et al. (2019) Table 3, last column:
> - ImageNet+DINOv2 ($S^{k=10}=1.4$) would be the 23rd/50 most affected dataset they report.
> - CIFAR10+DINOv2 ($S^{k=10}=2.6$) which has 10 dense clusters and is the 6th/17 least affected dataset-embedding combination we report in Table 4 would be 10th/50 most affected.
>
> These results are on the worse side, but remain quite typical.
> ## Crossover regime
> >Clarify crossover regime
>
> Please look at our answer to reviewer BTjk, marked with ## Crossover regime.
> ## Wording changes
> >W2 L220 & L227 + Q3 L234, 236 & Fig 1
>
> - L220: Will rephrase "spatial relationship" to "maintain the true local semantic structure […] removing only artifacts created by hubness."
> - L227: Will rephrase to ensure independence: "a given generated sample falls into the same real balls regardless of the other generated samples evaluated alongside it." (Again, no samples are removed).
> - L234, L236: We will insert "relatively" before density.
> - Fig 1: Will specify points are "uniformly drawn" from their mixtures.
> ## References not in the paper
> Komarov, I., Dashti, A. and D'Souza, R. M.. "Fast k-NNG construction with GPU-based quick multi-select." *PloS one* 9.5, 2014

---

> > ### Author Rebuttal · Reviewer_9L4p · 2026-04-01
> >
> > I thank the authors for the rebuttal. It clears up confused points that I raised on my reviews.
> >
> > When reading your paper, I feel it is hard to separate the two different spatial spaces: original space and ICDM-transformed space where the distance between $x_i$ and $x_j$ are now scaled by $\delta_i$ and $\delta_j$ called ICDM-distance. After transforming the real set into ICDM space, the density of real points (measure by avg kNN-ICDM-distance) looks more uniform, and this is the key property to separate between real and generated points used in the paper.
> >
> > __W1. Novelty of GICDM__
> >
> > Unfortunately, I feel GICDM's novelty is limited regarding to ICDM. I my view, GICDM and ICDM consider the scale $\delta_i$ computed on the dataset as a new dissimilarity (called ICDM distance). If we consider the generated points as adversarial points and real points as non-adversarial points, then the idea of using real-real in GICDM and computing distance between $x^g$ vs. real set  is quite neutral.
> > However, the time complexity is still O(tn^2 + nm) where n is the size of real set and m is the size of generated set. This limits the method on large-scale dataset.
> >
> > __W2) Unclear/superficial statements on key desiderate for hubness mitigation__
> >
> > I feel the statements on the key desiderata for hubness mitigation (L216-228) mix up the originial space and ICDM-transformed space. Indeed, the paper's organization makes me harder to follow the flow.
> >
> > __W3) Potential issues on the use of GICDM and proposed methodology__
> >
> > The assumption $\mu_i^{r, eq} \approx \bar{\mu}_i^{r, eq}$ for all $x_i$ looks quite strong to me as ICDM optimize $\sum_i$, not individual points. Since you run $t = 10$, you might want to verify the convergence of $\mu_i^{r, eq}$.
> >
> > __Overall__
> >
> > I have raised the overal recommendation from reject to weak reject but reduced my confidence to 3. While I understand well the contribution in the algorithmic way, I could not see the novelty of the work with respect to generative model evaluation problem and how important it is. I would prefer the discussion with other reviewers for the final decision.
> >
> > I also changed Soundness: 3 (1), Presentation 2 (2), Significant 2 (1), Originality 2 (1) where the value in bracket is before rebuttal.

---

> > > ### Author Response · Authors · 2026-04-03
> > >
> > > Thank you increasing your score and for the time you spent reviewing our paper.
> > > >W1. Novelty of GICDM
> > > >Unfortunately, I feel GICDM's novelty is limited regarding to ICDM. I my view, GICDM and ICDM consider the scale $\delta_i$ computed on the dataset as a new dissimilarity (called ICDM distance). If we consider the generated points as adversarial points and real points as non-adversarial points, then the idea of using real-real in GICDM and computing distance between $x^g$ vs. real set is quite neutral. However, the time complexity is still O(tn^2 + nm) where n is the size of real set and m is the size of generated set. This limits the method on large-scale dataset.
> > >
> > > We understand the first part as: why not use ICDM dissimilarities for real-real and use Euclidean distances for real-generated?
> > >
> > > The drawback of this approach would be that distance-based metrics compare real-real and real-generated distances when checking whether a generated sample is in a given real ball (whose radius is a real-real distance). If real-real distances are scaled by ICDM but real-generated distances are not, they cannot be meaningfully compared. This is where GICDM is needed: to also scale real-generated distances. GICDM ensures that the generated points map into the ICDM-scaled real space without distorting the underlying real manifold or interacting with each other.
> > >
> > > For the second part about the complexity and running times, please look at our last answer to reviewer 6sH2, marked with "## Scaling of GICDM". In short: it is not as bad as it seems.
> > >
> > > ## Two spaces
> > > >When reading your paper, I feel it is hard to separate the two different spatial spaces: original space and ICDM-transformed space where the distance between $x_i$ and $x_j$ are now scaled by $\delta_i$ and $\delta_j$ called ICDM-distance. After transforming the real set into ICDM space, the density of real points (measure by avg kNN-ICDM-distance) looks more uniform, and this is the key property to separate between real and generated points used in the paper.
> > > >
> > > >W2) Unclear/superficial statements on key desiderate for hubness mitigation
> > > >I feel the statements on the key desiderata for hubness mitigation (L216-228) mix up the originial space and ICDM-transformed space. Indeed, the paper's organization makes me harder to follow the flow.
> > >
> > > Thank you for this constructive feedback. We will clarify the desiderata further (changes from the pre-rebuttal version in bold):
> > >
> > > L216-228:
> > > >>1. Reduce hubness in the real dataset: **In the transformed distance space**, the local density of the real data manifold should be uniform, so that nearest neighbor relationships are meaningful and symmetric.
> > > >>2. Preserve the relative positioning of generated points: Hubness mitigation should maintain the **true local semantic structure** between generated points and the real data manifold, **removing only the artifacts created by hubness**.
> > > >>3. Ensure independence between generated points: The **distance scaling** of each generated point should depend only on its relationship to the real set **in the original space,** so that **a given** generated sample **falls into the same real balls regardless of the other generated samples evaluated alongside it.**
> > >
> > > ## Convergence
> > > >W3) Potential issues on the use of GICDM and proposed methodology
> > > >
> > > >The assumption $\mu_i^{r,eq} \approx \bar{\mu}_i^{r,eq}$ for all $x_i$ looks quite strong to me as ICDM optimize $\sum_i$, not individual points. Since you run $t=10$, you might want to verify the convergence of $\mu_i^{r,eq}$.
> > >
> > > Verifying the pointwise convergence is indeed crucial for justifying our method's assumptions.
> > >
> > > We took our 16 combinations of embedders (DINOv2, DINOv3, Inceptionv3, VGG16) and datasets (CIFAR-10, ImageNet, FFHQ, LSUN-bedroom) with K=10 and K=100 (the two values that are used to run metrics with K=5), and monitored the relative difference between individual $\mu_i^{r}$ and their mean $\bar{\mu}^{r}$.
> > >
> > > You can find the results [here](https://ibb.co/8Lxnhv96). Above the results at iterations 1, 5, 10, 15 and 20 we wrote the biggest relative difference. In all tested scenarii, all $\mu_i^r$ values are included in:
> > > - in [0.84, 1.51] after iteration 1 (=NICDN, the non-iterative version),
> > > - in [0.985, 1.022] after iteration 5,
> > > - in [0.9989, 1.0017] after iteration 10 (the value we stop at in our experiments),
> > > - in [0.99983, 1.00018] after iteration 15,
> > > - in [0.999966, 1.000036] after iteration 20.
> > >
> > > As these results show, after 10 iterations, the maximum deviation of any individual $\mu_i^r$  from the mean is less than 0.17% across all 16 diverse scenarios, empirically validating that ICDM successfully uniformizes the density for every individual point, rather than just minimizing the sum in aggregate. Therefore, our assumption that $\mu_i^{r,eq} \approx \bar{\mu}^{r,eq}$ for all $x_i$ holds in practice, and stopping at $t=10$ ensures convergence for GICDM.
> > >
> > > We will include this valuable check along with the plot in the Appendix.

---

### Official Review · Reviewer_BTjk · 2026-03-12

**Soundness:** 2
**Presentation:** 2
**Significance:** 3
**Originality:** 3
**Overall Recommendation:** 4
**Confidence:** 3

**Summary:**

Hubness is a property of high-dimensional datasets wherein a small collection of points (the “hubs”) appear disproportionately often in the set of nearest neighbors to other points. In general, hubness can be considered a challenge for high-dimensional data analysis because it can confuse the “signal” encoded in the nearest neighbor structure.

This paper argues that existing metrics for generative model evaluation (i.e., measuring how “close” a generative model is to its corresponding real data distribution, based on finite samples from both distributions) are distorted due to the hubness phenomenon. This paper outlines an approach towards reducing hubness for these purposes.

Existing methods for hubness mitigation, like Iterative Contextual Dissimilarity Measure (ICDM), take in a single dataset’s distance matrix and output a new distance matrix with the hubs suppressed. This paper adapts the ICDM technique for de-hubbing the distances of both a dataset and its synthetic counterpart; they call their method generative ICDM, or GICDM. The authors show that GICDM as a preprocessing step improves the performance of existing methods for generative model evaluation.

**Compliance With Llm Reviewing Policy:**

Affirmed.

**Final Justification:**

I have increased my score because the rebuttals have more or less clarified their characterization of hubness as a problem in this context. I still think the treatment and discussion of hubness could be more "balanced"; for instance, I am still unsure about it being called a "mathematical artifact." I wouldn't call the concentration of measure phenomenon a mathematical artifact.

I have also increased my "significance" score because I recognize that the apparent success of GICDM in the experiments is promising and this.

The "presentation" score remains the lowest.

**Key Questions For Authors:**

(re: Figure 2 and Section 2.1) Fix dimension d and neighborhood size k. Draw n samples from an isotropic Gaussian in d-dimensions. What is (roughly) the growth rate of max_x O_k(x) as a function of n,k,d? More generally, what can we say about the number of samples needed for a significant hubness phenomenon to emerge? If there is prior work on this it would be great to discuss it.

(re: general + Appendix A.1) In real datasets what do the “hubs” look like? For instance, for ImageNet DinoV2 embeddings, discussed in line 689.

If hubs are typically a small fraction of the dataset, why not just remove them from the dataset? How does this perform relative to the GICDM pre-processing step? This would be a good experiment to run and discuss, as it makes for a significantly simpler approach to hubness reduction.

**Limitations:**

yes

**Strengths And Weaknesses:**

**Strengths**:
(re: Significance/Originality)'

(1) Generative model evaluation is important. This paper seems to make progress on improving the general understanding of challenges for generative model evaluation, by highlighting hubness. It also improves the toolkit for generative model evaluation, in that the pre-processing done by GICDM seems to improve the performance of existing generative model evaluations.
(re: Presentation)

(2) I think the presentation and toy demonstrations of hubness are quite helpful. Figure 2 and 4 in particular get the message across quite cleanly.

**Weaknesses**:
(re: Significance/Soundness)

(1) The main claim of this paper is that, in the context of generative model evaluation,
GICDM( (real dataset, synthetic dataset) ) = (modified real dataset, modified synthetic dataset)
is better to work with than (real dataset, synthetic dataset). I think the paper could do more to address the potential concerns that (a) the modified datasets have potentially lost important structure and (b) removing hubness “hacks” the flawed performance metrics of fidelity and coverage but may not actually be desirable. This concern is compounded by (c) the relatively heuristic “derivation” of GICDM presented in Section 4.2 and (d) the relatively brief discussion of empirics on the last page of the paper.

(re: Presentation)

(2) I have trouble with the framing of the paper around the idea that hubness is inherently a problem that needs to be fixed (e.g. the discussion in lines 88-92 right column). Hubness is perhaps better understood as a challenge associated with analyzing high-dimensional data, in that it complicates our usual understanding of neighborhoods as symmetric relationships. It should be emphasized that removing hubs or otherwise “de-hubbing” the distances in a dataset could potentially destroy useful information in the data. Figures 4c and 4d show this to some extent.

(3) I am generally confused by Section 4.4: specifically the discussion of “crossover dimension” and Proposition 4.3. It feels out of place. I think it needs more context.

(4) I do not think Proposition 4.3 belongs in the main text at all.

(5) I think more space should be dedicated to more discussion of experiments; I am not sure the one page at the end does it justice.

(6) The use of “approx” signs in proofs and theorem-like statements (on page 5, for instance) is in my opinion unacceptable. It would be more acceptable in a “Proof Sketch.”

(7) I think it is a bit unusual for the related work section to appear after the presentation of the method.

---

> ### Author Rebuttal · Authors · 2026-03-30
>
> Thank you for the positive feedback on the motivation and hubness presentation.
> ## Validating GICDM
> >W1, W2 end: structure loss? W1d, W5: experiments details
>
> Guaranteeing that metrics capture all aspects is difficult. This concern appears when we work in embedding spaces that might lose structure. The standard validation approach is extensive empirical testing, notably correlation with human judgment.
>
> We compare >10 metrics on 17 synthetic tests (Apdx.C), 42 sets for human correlation with 2 embeddings (Apdx.D), and 3 real-data tests (Apdx.E). GICDM consistently improves performance. We’ll expand Sec. Experiment, to note, e.g., that GICDM fixes issues while preserving already satisfied criteria.
> ## Hubness should be corrected
> >W2 Fix hubness?
>
> Distance-based evaluation assumes feature space distances are semantically meaningful. If $x$ has $y$ as a nearest neighbor (NN) but $y$ doesn’t have $x$ due to hundreds of far closer samples, this one-way link is a **mathematical artifact** of hubness, **not a true semantic link**.
>
> See [Jégou et al. (2010) Figure 2](https://inria.hal.science/inria-00439311v3) for a visual illustration: the only semanticly meaningful NNs of the query image are reciprocal. Forcing symmetry prunes these non-meaningful artifacts.
>
> Using NN classification "to measure semantic correctness", Feldbauer & Flexer (2019) show that scaling methods restoring NN symmetry "consistently reduce hubness and often improve classification performance" across 50 datasets "without any noticeable risk of performance degradation." We will clarify this L181-184L.
> ## When does hubness become an issue
> >Q1: Growth rate of max_x O_k(x) + from which n is hubness an issue?
>
> To the best of our knowledge, there is no known formula for this growth rate. But, **hubness is an inherent property of data distributions** in high dimensions (Radovanovic et al., 2010). While n affects $\max_x O_k(x)$ and hubness estimates, **the true hubness level is inherent to the distribution** (with $d$, $k$).
>
> Feldbauer et al. (2018, Fig.1b) detect hubness with just 100 samples. In high-dimensional Gaussians, central samples become hubs **for any $n$**.
> ## Removing hubs
> >Q3: Just remove hubs?
>
> - **Removing hubs creates new ones**. Removing current hubs, which sit in the densest regions, makes points in the next highest-density region hubs. E.g., for CIFAR-10/DINOv2 (6th least hubness-affected out of our 17 sets, Table 4), removing the 10% most-occurring samples only shifts $h_1^5(1\\%)$ from $6.5$ to $4.39$, and the antihub proportion ($A^5$) from 14% to 12%,, which is still worse than ImageNet/DINOv2. (Will clarify L142).
> - **Antihubs are problematic blind spots**. Antihubs ($O_k(x)=0$), have no real ball around them. Nearby generated samples go unevaluated: fidelity $=0$ could mean a bad generated sample or a generated antihub. (Will emphasize L69-74L, L83-86R, L142, L352R).
> ## Crossover regime
> >W3, W4 The crossover regime lacks context.
>
> We will clarify Section 4.3 works only outside the crossover regime L310-311L:
> >>If a generated point is not well aligned with the real data manifold **and we are outside the crossover regime (discussed in the next section)**, its distances to its real neighbors […] will not be consistent with those observed among real points.
>
> Section 4.4 will clarify that poorly aligned points can have realistic NN distances if the local density hits the *crossover regime* (dependent on density, $N$, $d$, $K$).
>
> L321R will note that while quantified for Gaussians, the mechanism is general: neighborhood scale scales with $K$. Changing $K$ shifts the crossover regime where out-of-manifold distances coincidentally match in-manifold ones. Using two sufficiently different $K$'s (e.g., $K_2 = 10 K_1$) prevents this setup. Fig. 6 shows that without multiscale filtering (green), disjoint distributions erroneously get non-zero scores here.
> ## Approx signs
> >W6 Remove $\approx$
>
> We revised Proposition 4.1's proof using strict equalities, replacing $p$ with $\hat{p}\_{KNN}$ (L258) and $\hat{p}\_{kNN}$ (L261, 264). Substituting the resulting expression for $\mu_i$ yields $\hat{p}\_{\mu, K}(x_i) = (\sum_{k=1}^K w_k \hat{p}\_{\text{k-NN}}(x_i)^{-1/d})^{-d}$ with $w_k = k^{1/d}/\sum_{j=1}^K j^{1/d}$. By the Silverman-Toeplitz theorem (Toeplitz, 1911) and continuity, $\hat{p}_{\mu, K}(x_i) \xrightarrow{P} p$.
>
> We reframed Corollary 4.2 (L220-221):
> >>At ICDM convergence, **when $\forall i, |\mu_i - \bar{\mu}| < \epsilon$ for an arbitrarily small $\epsilon > 0$**, the local density estimates $\hat{p}_{\mu, K}(x_i)$ are equal for all $x_i$.
>
> (L258R, L264R updated similarly).
> ## Presentation & Hub Examples
> >W7 & Q2
>
> We will move related work before "GICDM".
> Q2: The 64 top hubs in ImageNet/DINOv2 are [here](https://ibb.co/zHN3cWmK). They seem to lack common visual traits and appear random.
> ## References not in the paper
> Toeplitz, O. Über allgemeine lineare mittelbildungen. *Pracematematyczno-fizyczne*, 22(1):113–119, 1911.

---

> > ### Author Rebuttal · Reviewer_BTjk · 2026-04-03
> >
> > Thank you for the detailed and thoughtful rebuttal.
> >
> > My main remaining qualm is the characterization of hubness in the paper. I understand that there is significant evidence (from Jegou, Feldbauer and Flexer, as well as your own paper) that in real-world datasets, reducing hubness improves downstream performance on classification/clustering tasks. However, I still disagree with the very general characterization given in this rebuttal (representative of the tone of the paper) that hubness is inherently a issue. Even Radovanonic 2011 doesn't rule out the idea that hubs could provide useful information: they write that "identifying hubness within data and methods from other fields can be considered an important aspect of future work, as well as designing application-specific methods to mitigate or **take advantage of the phenomenon**."
> >
> > On a related note, I echo Reviewer 6sH2's concern re: "Loss of information after removing hubness." While it is promising that ClippedDensity and ClippedCoverage, it would be good to sweep over more success metrics (especially since (Raisa 2025), one of the more important citations of this paper, argues that "all existing fidelity and diversity metrics are flawed").

---

> > > ### Author Response · Authors · 2026-04-03
> > >
> > > Thank you for your thoughtful comments.
> > > ## Characterization of hubness
> > > >My main remaining qualm is the characterization of hubness in the paper. I understand that there is significant evidence (from Jegou, Feldbauer and Flexer, as well as your own paper) that in real-world datasets, reducing hubness improves downstream performance on classification/clustering tasks. However, I still disagree with the very general characterization given in this rebuttal (representative of the tone of the paper) that hubness is inherently a issue. Even Radovanonic 2011 doesn't rule out the idea that hubs could provide useful information: they write that "identifying hubness within data and methods from other fields can be considered an important aspect of future work, as well as designing application-specific methods to mitigate or **take advantage of the phenomenon**."
> > >
> > > We agree that hubs can sometimes represent a valid signal (e.g. in graph data). We do not wish to imply that hubs are universally bad, but rather that they are non meaningful in our specific context of localized, semantic distance-based tasks (like k-NN classification). In these specific scenarios, if $x$ has $y$ as a nearest neighbor, but $y$ does not have $x$ even as a 1000th nearest neighbor, then this relationship is not likely to be semanticaly meaningful.
> > >
> > > We will emphasize in the text that claims regarding the necessity of mitigating hubness are specific to our context e.g. L181-184 left (changes in bold):
> > > >>[…] while the others are scaling methods that seek to restore symmetry in nearest neighbor relationships by scaling distances. **Scaling methods that force symmetry consistently increase nearest neighbor classification accuracy (Feldbauer & Flexer, 2019), indicating that they improve the semantic correctness of local neighbor relationships by reducing the influence of one-way relationships induced by hubness.** The Gaussian variant of Mutual Proximity […]
> > > ## Testing many metrics
> > > >On a related note, I echo Reviewer 6sH2's concern re: "Loss of information after removing hubness." While it is promising that ClippedDensity and ClippedCoverage, it would be good to sweep over more success metrics (especially since (Raisa 2025), one of the more important citations of this paper, argues that "all existing fidelity and diversity metrics are flawed").
> > >
> > > We completely agree that evaluating many metrics is essential. We acknowledge that the main text gave the impression that we only evaluated with Clipped Density and Clipped Coverage. Because of space constraints, the main body highlighted only these two as they perform best. However, we actually tested more than 10 different metrics (in Appendix C, D, and E).
> > >
> > > In particular, when running the full Räisä 2025 benchmark (Appendix C, Tables 6 and 7), the improvements brought by GICDM are not limited to those two metrics. We found (L947-948) that "GICDM consistently improves Purpose results (shape of the curve) on both the ”Gaussian Std. Deviation Difference” and ”Hypersphere Surface” tests.".
> > >
> > > Further following your request for a more detailed Experiments section, we have revised the main text L416-422 left to:
> > > >>Synthetic Benchmark. We evaluated standard metrics and their GICDM-corrected versions on the **extensive** synthetic benchmark **proposed by**  Räisä et al. (2025). **This benchmark evaluates the evolution of metric results across 14 *Purpose* scenarios, tests whether metrics yield the expected values (e.g., 1 for realistic generated data) in 13 *Bounds* cases, and checks 3 *Other* miscellaneous tests.** **We ran this benchmark on more than 10 metrics (with full details in Appendix C), and found that GICDM consistently improves two Purpose tests.** Table 1 **summarizes the results for the best-performing fidelity and coverage metrics: applying** GICDM **substantially** improves performance, **passing 23 criteria instead of 18 for Clipped Density with GICDM, and 22 instead of 18 for Clipped Coverage with GICDM. This underlines** that hubness **is** indeed the root cause of several failures **and that GICDM successfully corrects this issue without altering performance on criteria that were already satisfied.**

---

### Decision · Program_Chairs · 2026-04-30

**Decision:**

Accept (regular)

**Comment:**

The paper suggests an approach to mitigate hubness is high-dimensional vector embeddings for the purpose of evaluating generative models. Such models are often evaluated by how similar the distribution of generated samples is to the distribution of real samples. This comparison is typically kNN-based, and can be affected by hubness. The authors adapt an existing iterative procedure to mitigate hubness for this specific use-case.

The reviewers appreciated that the paper studies an interesting question and that the results are compelling. Some of the reviewers had doubts about the sufficient novelty, but in discussion the consensus was that applying ICDM to this problem is sufficiently novel, and I agree. Given somewhat mixed opinions of the reviewers, I recommend "weak acceptance".

One concern that reviewers had was related to phrasing that can be understood as "removing hubness is essential". The reviewers thought it is fairer to say that "removing hubness can help certain tasks" (but maybe not all tasks; and removing hubness can also have potential detrimental side-effects). I encourage the authors to rephrase the paper accordingly, and maybe also have some explicit discussion about potential side-effects.

I agree with the reviewers that the flow of the sections can be improved, Related Work should be moved to the beginning, etc.

One final comment from my side is that the desiderata in Section 4 should be better motivated. To me it is not obvious that each generated point should be treated independently. If the idea behind evaluating a generative model is to assess how close the SET of generated samples is to the SET of real samples, then why cannot we treat generated samples as a SET rather than individually? I feel this should be better motivated, as these desiderata are crucial for your GICDM.